# Structural basis of lipid-linked galactan export by the mycobacterial ABC transporter Wzm-Wzt

Alisa A. Garaeva [1,8] ✉, Viktória Fabianová [2,8], Karin Savková[2,8], Stanislav Huszár [2], Xiaochao Xue[3,7], Todd L. Lowary [3,4,5], Katarína Mikušová [2] ✉ & Markus A. Seeger [1,6] ✉

Mycobacteria, including *Mycobacterium tuberculosis*, possess a unique cell envelope containing arabinogalactan, a heteropolysaccharide critical for cell wall integrity and target of several tuberculosis drugs. The cytosolic precursor of arabinogalactan, lipid-linked galactan (LLG), is translocated across the plasma membrane by the essential ABC transporter Wzm-Wzt through a molecular mechanism that is poorly understood. Here, we present a series of cryo-EM structures of Wzm-Wzt from *Mycobacterium abscessus*, representing different conformations of the transport cycle. Conserved residues lining the proposed LLG translocation pathway were investigated by three orthologous functional assays, revealing that the cytosolic gate helix (GH) plays a key functional role in polysaccharide transport. Our data suggests that the hydrophobic polyprenyl-moiety is translocated first, followed by the galactan-polysaccharide, which requires Wzm-Wzt to open a continuous channel through which the sugar chain is ratcheted at the expense of ATP hydrolysis. Our results provide a rational basis for the development of drugs that inhibit mycobacterial cell wall biosynthesis.

Cell envelopes of both Gram-positive and Gram-negative bacteria are decorated with a highly variable coat of structurally diverse glyco-conjugates. In case of pathogens, these macromolecules enable them to interact with the host's immune system, to contribute to biofilm formation or virulence, in addition to maintaining the cell wall integrity[1,2]. Mycobacteria, which include pathogens causing tuberculosis (*Mycobacterium tuberculosis*), leprosy (*M. leprae*) or other serious infections of skin or soft tissues (nontuberculous mycobacteria, NTMs), differ from classical Gram-positive and Gram-negative bacteria by the distinct cell envelope, as well as by the presence of unique cell wall glycoconjugates[3]. Among these, the heteropolysaccharide

arabinogalactan, which links the peptidoglycan layer with the myco-bacterial outer membrane (also called "mycomembrane"), plays an essential role in preserving the structural integrity of the cell wall. Notably, several drugs used in tuberculosis treatment or in clinical development, including ethambutol or benzothiazinones, target ara-binogalactan biosynthesis[4].

The biosynthesis of arabinogalactan takes place in two cellular compartments: the cytosol and the periplasm. The galactan component is produced in the cytoplasm as a lipid linked oligo-saccharide (decaprenyl pyrophosphate *N*-acetylglucosaminyl-rham-nosyl-galactofuranose$_{-20-30}$; decaprenyl-*P*-*P*-Glc*p*NAc-Rha*p*-Gal*f*$_{-20-30}$)

[1]Institute of Medical Microbiology, University of Zurich, Zurich, Switzerland. [2]Department of Biochemistry, Faculty of Natural Sciences, Comenius University in Bratislava, Bratislava, Slovakia. [3]Department of Chemistry, University of Alberta, Edmonton, AB, Canada. [4]Institute of Biological Chemistry, Academia Sinica, Taipei, Taiwan. [5]Institute of Biochemical Sciences, National Taiwan University, Taipei, Taiwan. [6]National Center for Mycobacteria, University of Zurich, Zurich, Switzerland. [7]Present address: The Institute for Molecular and Cellular Therapeutics, Chinese Institutes for Medical Research, Beijing, China. [8]These authors contributed equally: Alisa A. Garaeva, Viktória Fabianová, Karin Savková. ✉e-mail: alisa.garaeva@uzh.ch; katarina.mikusova@uniba.sk; m.seeger@imm.uzh.ch

by the stepwise action of membrane embedded (WecA) and membrane associated glycosyl transferases (WbbL, GlfT1, GlfT2)[5–7]. The translocation of this lipid-linked galactan (LLG) across the plasma membrane is mediated by the ATP-binding cassette (ABC) transporter Wzm-Wzt, which we recently identified and functionally characterized[8]. Arabinosyltransferases acting at the periplasm then attach arabinosyl residues on the galactan chain of LLG[9]. The final steps in synthesis of the cell wall core include joining arabinogalactan with peptidoglycan by ligases from the LytR-CpsA-Psr (LCP) family[10,11] and the attachment of mycolic acids catalyzed by mycolyl transferases from the Ag85 family[12].

All ABC transporters minimally consist of two nucleotide binding domains (NBDs) responsible for ATP binding and hydrolysis to energize the active transport across the membrane mediated by the transmembrane domains (TMDs). Wzm-Wzt belongs to the type V subfamily of ABC transporters[13], which is involved in the export of various substrates, including drugs and sterols[14–16]. In mycobacteria, the gene region encoding Wzm-Wzt has an unusual organization, because the genes encoding Wzm (the TMD) and Wzt (the NBD) are separated by the gene encoding the initiating galactosyltransferase GlfT1[8,17]. Only three representatives of type V ABC transporters involved in polysaccharide export were thus far structurally characterized: (i) Wzm-Wzt from the Gram-negative bacterium *Aquifex aeolicus* (Wzm-Wzt$_{Aa}$), which transports a specific O-antigen to the periplasm before its attachment to lipid A to form a mature lipopolysaccharide[18–21]; (ii) TarGH from the Gram-positive bacteria *Alicyclobacillus herbarius* and *Staphylococcus aureus*, which exports wall teichoic acids (WTA) for decoration of peptidoglycan[22,23], and (iii) KpsMT from *Schlegelella thermodepolymerans*, involved in the translocation of capsular polysaccharides (CPS) across the Gram-negative cell envelope[24].

Similar to the galactan ABC transporter from mycobacteria, both Wzm-Wzt$_{Aa}$ and TarGH export polyprenyl-diphosphate-linked polysaccharide substrates. In contrast, the KpsMT substrate is anchored to a phosphatidylglycerol. Although these structurally characterized transporters share several common features, they also exhibit notable differences. For instance, Wzm-Wzt$_{Aa}$ possesses an extra carbohydrate binding domain (CBD), which is critical for recognizing the sugar moiety of the substrate – a feature absent in other family members. In addition to the conserved sequence motifs in the NBDs common to all ABC transporters, Wzm-Wzt and TarGH feature a gate helix (GH)[19] or a gate loop (GL)[22] (Supplementary Fig. 1), respectively, which is inserted into the NBD sequence and has been proposed to play a role in substrate engagement. However, functional evidence supporting this role is still lacking. Unlike Wzm-Wzt and TarGH, KpsMT lacks the gate helix[24]. A conserved element common to all three transporters is a short loop containing leucine followed by glycine, which joins an N-terminal interface helix and the first transmembrane helix of TMD (Supplementary Fig. 1). It is termed the LG loop in Wzm-Wzt$_{Aa}$[19] and KpsMT[24] or IF2-TM1 loop in TarGH[22]. It was hypothesized that conserved hydrophobic residues within the LG-loop could be involved in the translocation of the polysaccharide substrate across the TMD channel[19]. Recently, W45 in the LG-loop of KpsM was shown to be essential for CPS transport[24].

The chemical diversity and the amphipathic biophysical properties of lipid-linked polysaccharide substrates as well as the lack of sensitive functional assays to study site-directed mutations explain why the transport mechanism of these ABC transporters is still poorly understood. To date, no cryo-EM structure featuring a polyprenyl-diphosphate-linked polysaccharide substrate within the TMD channel of Wzm-Wzt$_{Aa}$ or TarGH has been reported. In contrast, recent cryo-EM studies of glycolipid-bound KpsMT provided structural insights into phosphatidylglycerol-linked Kdo (3-deoxy-D-manno-oct-2-ulosonic acid) monosaccharide binding within the TMD translocation channel[24].

To shed light on the molecular mechanism of LLG transport, here we show cryo-EM structures of the arabinogalactan exporter from the opportunistic pathogen *Mycobacterium abscessus* in multiple conformations, both in detergent and nanodiscs as well as in the presence of an LLGanalog.

## Results

### Cryo-EM structures of Wzm-Wzt from *Mycobacterium abscessus* in LMNG

In an expression and purification screen of Wzm-Wzt homologues from different mycobacterial species (Supplementary Fig. 1), we identified Wzm-Wzt from *Mycobacterium abscessus* (Wzm-Wzt$_{Mabs}$) as the most suitable candidate for structural analysis. The protein complex was expressed from a pET-Duet vector in *E. coli* with an N-terminal 6-His tag on Wzm (i.e. the TMD) and was subsequently purified in the detergent LMNG (see methods). Next, we determined cryo-EM structures of wild-type Wzm-Wzt$_{Mabs}$ either in the absence (apo) or the presence of ATP-Mg, the latter corresponding to ATP turnover conditions (Fig. 1, Supplementary Fig. 2–4, Supplementary Table 1). Previous studies identified farnesyl-*P-P*-Glc*p*NAc-Rha*p*-Gal*f*-Gal*f*, a truncated version of the natural substrate decaprenyl-*P-P*-Glc*p*NAc-Rha*p*-Gal*f*-Gal*f*, as an efficient acceptor for galactan polymerization by GlfT2[25]. We anticipated that this substrate analog could also be recognized by Wzm-Wzt$_{Mabs}$, because it possesses the Glc*p*NAc pyrophosphate group, which was shown to be critical for the translocation of WTA precursor by TarGH[26]. Therefore, farnesyl-*P-P*-Glc*p*NAc-Rha*p*-Gal*f*-Gal*f* was added to most preparations analyzed by cryo-EM (Supplementary Table 1), but we only observed unassigned density consistent with an amphipathic molecule in the binding cavity when the transporter was prepared in nanodiscs in the presence of the substrate analog (see below).

The molecular assembly of Wzm-Wzt$_{Mabs}$ is analogous to that of Wzm-Wzt$_{Aa}$; the TMDs (Wzm) each encompass six transmembrane helices (TMs) and homodimerize via extensive protein-protein interactions. The NBDs (Wzt) are connected to the TMDs via coupling helices. Hence, the fully assembled export machinery consists of four separate protein chains, two identical TMDs and two identical NBDs. In the absence of added ATP-Mg, the NBDs of Wzm-Wzt$_{Mabs}$ are disengaged and the TMDs form an inward-facing cavity extending to the middle of the membrane (Fig. 1a). For the ATP turnover dataset, we observed two distinct conformations. The first conformation has disengaged NBDs with two ADP bound, which is identical to the apo conformation (RMSD 0.404 Å). The second conformation has dimerized NBDs with two ATP-Mg molecules sandwiched at the NBD interface, while the TMDs circumference a cavity that extends further up towards the periplasm and is constricted, but not entirely closed, at the cytosolic end (Fig. 1a).

We also determined a cryo-EM structure of Wzm-Wzt$_{Mabs}$ containing the E178Q mutation in the Walker B motif of the NBD, which traps ATP-Mg at the NBDs (Fig. 1a, b, Supplementary Fig. 5, Supplementary Table 1). The resulting structure is identical to the ATP-Mg bound wild-type transporter (RMSD 0.478 Å) (Fig. 1a).

### Structural analysis of the TMD region

Wzm-Wzt$_{Mabs}$ contains structural elements characteristic for type V ABC transporters involved in lipid-linked polysaccharide export (Fig. 1b, d). The polypeptide chain of Wzm$_{Mabs}$ (the TMD) starts with a broken interface helix that is positioned parallel to the membrane plane, followed by six transmembrane helices (TM1-6). The Wzm-Wzt$_{Mabs}$ (i.e. the TMD-NBD) interface has an extensive buried surface area of 1655 Å$^2$ (interface of chain A and chain C of apo structure calculated by PDBePisa). The TMD-NBD interface is mainly established by the TM2-TM3 loop forming a coupling helix and a short helix following TM6 at the C-terminal end of the Wzm polypeptide (Fig. 1d). The second half of the N-terminal interface helix as well as the loop

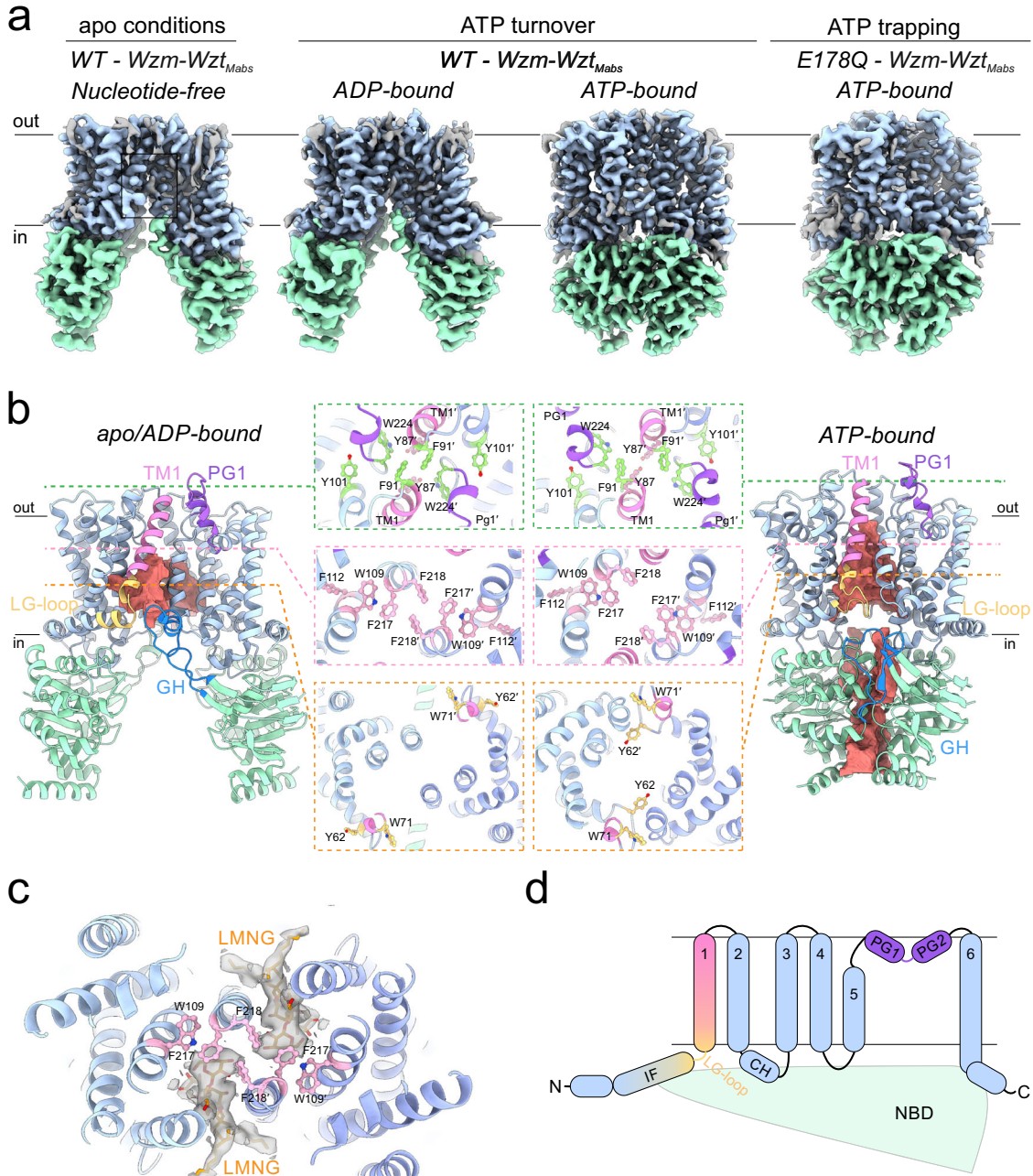

**Fig. 1 | Wzm-Wzt_{Mabs} structures in LMNG. a** Cryo-EM maps of Wzm-Wzt_{Mabs} in LMNG micelle. **b** Models of apo/ADP-bound and ATP-bound Wzm-Wzt_{Mabs} with labelled structural elements are shown as cartoon. Slices at three levels indicated by green, pink and orange dotted lines represent views from the periplasm on aromatic belts within the TMD. Cavities calculated with the 3 V tool are shown in red.

**c** LMNG binding site (orange sticks) with corresponding cryo-EM densities (shown as grey transparent surface at 5σ) between two adjacent TMDs of Wzm-Wzt_{Mabs}. Residues surrounding LMNG are shown as pink sticks. **d** Topology of Wzm-Wzt_{Mabs} TMD. The NBD is shown schematically. IF – interface helix, CH – coupling helix, PG – periplasmic gate helix.

connecting TM4 and TM5 are involved in additional interactions. The loop connecting the interface helix with TM1 is referred to as the LG-loop and has been proposed to be involved in regulating substrate entry into the transporter[19]. TM5 is shorter than the other TM helices and forms a loop at its periplasmic end, followed by the periplasmic gate helices 1 and 2 (PG1-2), which form a V-shaped structure pointing toward the center of the TMD. TM5 and PG1 have been previously suggested to orchestrate the substrate's lateral exit[20].

The TMDs have three layers of aromatic residues lining the potential substrate translocation channel (Fig. 1b). The periplasmic aromatic belt includes F91^{Wzm}, Y101^{Wzm} and W224^{Wzm}, whose side chain positions re-orient in response to NBD closure as the transporter

transits from its ADP-bound to its ATP-bound conformation (Fig. 1b). W224^{Wzm} is part of PG1, which moves away from the central symmetry axis during the conformational transition, thereby initiating the opening of the extracellular gate. Other notable conformational changes at the periplasmic gate involve the loop connecting TM5 and PG1 (Supplementary Fig. 6a, b). A second belt, composed of Y87^{Wzm}, W109^{Wzm}, F112^{Wzm}, F217^{Wzm} and F218^{Wzm}, forms a central aromatic layer (Fig. 1b). This belt constricts the inward-facing cavity of the ADP-bound transporter, but opens up upon ATP-induced NBD closure resulting in a partial channel opening towards the periplasm. Finally, Y62^{Wzm} and W71^{Wzm}, which are part of the LG-loop form a third aromatic belt and surround the cytoplasmic channel entrance. Y62^{Wzm} undergoes a

drastic conformational change upon transition to the ATP-bound state and thereby constricts the cytosolic entrance (Fig. 1b).

## LMNG binds at the inward-facing cavity

In both the apo and in the ADP-Mg-bound structures, we observed densities for the detergent LMNG (Fig. 1c). Two symmetry-related molecules of LMNG intercalate into hydrophobic pockets, neutral in charge and formed at the interface of two TMD subunits, between TM1 and TM5. The position of LMNG overlaps with the glycolipids observed in the structure of KpsMT[24]. The sugar moiety of LMNG is coordinated by the central aromatic belt involving $W109^{Wzt}$ in TM2, $F217^{Wzt}$ in TM5 and $F218^{Wzt}$ in TM5 of the adjacent protomer. Although the LLG analog farnesyl-$P$-$P$-Glc$p$NAc-Rha$p$-Gal$f$-Gal$f$ was added to the ATP turnover sample, based on which the ADP-Mg-bound structure was determined (Supplementary Table 1), we did not observe density for it; rather the same density corresponding to LMNG was found both in the apo and the ADP-Mg-bound structures. Interestingly, in the ATP-Mg-bound structures of wild-type and $E178Q^{Wzt}$ Wzm-Wzt$_{Mabs}$, we did not observe density for LMNG in the cavity. This can be explained by the conformational changes that open up the central aromatic belt thereby breaking the interactions with LMNG's sugar moieties, or by the constriction of the periplasmic gate that might have squeezed out LMNG from the cavity.

## The cytosolic gate helix undergoes extensive conformational changes

As mentioned above, a peculiarity of type V ABC transporters involved in polyprenol-linked polysaccharide transport is a sequence insertion in the NBD right after the conserved A-loop tyrosine, named the gate helix (GH) (Supplementary Fig. 1). As opposed to previous structural analyses of Wzm-Wzt$_{Aa}$, we could resolve the conformational rearrangements of this distinct structural element of Wzm-Wzt$_{Mabs}$ in both conformations. In the ADP-Mg-bound structure having the NBDs widely apart, the GH forms an α-helix on two β-sheet "handles", and extends towards the membrane. Thereby, it reaches TM5 and the LG-loop of the adjacent protomer, approximately at the putative solvent-lipid boundary (Fig. 2a). This conformation of the GH is highly flexible, as indicated by the weak cryo-EM densities in this region (Supplementary Fig. 4). However, we were able to model the GH using maps processed with different B-factors for sharpening and low-pass filters (see Methods). The GH structure is similar to what was reported previously for Wzm-Wzt$_{Aa}$, but in Wzm-Wzt$_{Mabs}$ the GH reaches closer to the TMD, thereby constricting the cytosolic channel entrance (Fig. 2b, c). Remarkably, the GH undergoes extensive structural rearrangements when the NBDs close in response to ATP-Mg binding. The α-helix unfolds, and the GH moves perpendicularly relative to the membrane plane, thereby establishing an antiparallel β-strand to the existing A-loop β-sheet of the NBD (Figs. 1b, 2d, e). As a consequence, the backbone carbonyl of the GH-residue $G32^{Wzt}$ appears to establish a direct interaction with the ribose hydroxyl group of ATP (Fig. 2f).

## Establishment of functional assays to study Wzm-Wzt$_{Mabs}$ variants

We have recently demonstrated that mycobacterial Wzm-Wzt transports decaprenyl-$P$-$P$-Glc$p$NAc-Rha$p$-Gal$f_{-20-30}$[8]. With the aim to test the functional consequences of structure-guided point mutants of Wzm-Wzt$_{Mabs}$, we established three orthogonal functional assays in the model organism *Mycobacterium smegmatis*. The first assay builds on the growth defect if the *wzm-wzt* operon is knocked down in *M. smegmatis*, while the other two assays probe changes to the (glyco)-lipid composition. We used an anhydrotetracycline (ATc)- induced CRISPR interference (CRISPRi) strain of *M. smegmatis* targeting the *wzt* gene (*Msmeg* CRISPRi-*wzt*) (Supplementary Fig. 7a), which we previously demonstrated to exhibit severe growth attenuation and accumulation of LLG upon induction of repression[8]. A characteristic

phenotypic hallmark of *wzm-wzt* repression is the overproduction of extractable cell wall lipids, trehalose monomycolates (TMM) and trehalose dimycolates (TDM), which accumulate due to the lack of arabinogalactan serving as an acceptor of mycolic acids for the cell wall core construction[8] (Fig. 2g). We hypothesized that we can complement *Msmeg* CRISPRi-*wzt* grown in the presence of ATc with the *wzt-glfT1-wzm* operon from *M. abscessus* (*wzt-wzm*$_{Mabs}$) provided on a plasmid. The sequences of *wzt*$_{Msmeg}$ and *wzt*$_{Mabs}$ differ by three nucleotides in the region of the targeting sgRNA, hence the sgRNA is expected to only knock down the endogenous *wzt-glfT1-wzm* operon on the *M. smegmatis* genome (Supplementary Fig. 7a). For complementation, we constitutively expressed genes encoding for wild-type or mutated Wzm-Wzt$_{Mabs}$ in *M. smegmatis* using the replicative pFLAG vector[27]. When expressing the genes encoding for wild-type Wzm-Wzt$_{Mabs}$, growth of the knock-down strain in the presence of ATc was fully restored (Fig. 2h, Supplementary Fig. 7b, c). This shows that the Wzm-Wzt$_{Mabs}$ transporter is functional in *M. smegmatis*. As expected, the *M. abscessus wzt-wzm* operon transcript is not affected by the CRISPR interference targeting the *wzt* of *M. smegmatis*, a notion that was confirmed by gene expression analysis (Supplementary Fig. 7d).

To test the sensitivity of our functional assay, we complemented the *Msmeg* CRISPRi-*wzt* strain with Wzm-Wzt$_{Mabs}$ carrying either the catalytically-dead $E178Q^{Wzt}$ variant or the $F13A^{Wzt}$ variant in the NBD (Fig. 2f). $F13^{Wzt}$ is located in the A-loop preceding the GH and is responsible for aromatic stacking interactions with ATP's adenine moiety (Fig. 2f). In a deep mutational scanning study of the drug efflux pump EfrCD, the analogous mutation (Y359A in EfrD) resulted in a partial impairment of transport[28]. As expected, the $E178Q^{Wzt}$ variant was severely compromised in growth akin to the empty vector control and accumulated TDM and TMM, clearly showing that LLG transport requires the energy from ATP hydrolysis (Fig. 2h, i, Supplementary Fig. 7b, c and Supplementary Fig. 8a). By contrast, the $F13A^{Wzt}$ variant did not exhibit a growth deficit as compared to the wild-type transporter and did not show TDM and TMM accumulation (Fig. 2h, i, Supplementary Fig. 7b, c and Supplementary Fig. 8a), showing that these two functional assays are not suitable to monitor partial transport defects.

We therefore set out to establish a more sensitive assay that directly assesses LLG biosynthesis by performing metabolic labelling of the cultures with [14C]-glucose. This substrate is efficiently processed in mycobacteria, which results in incorporation of [14C]-label into cellular components, including the cell wall and its precursors. In this assay, we extracted LLG with organic solvent and monitored the amount of [14C]-galactose in the hydrolyzed samples, as described before[8]. Complementation with the catalytically-dead $E178Q^{Wzt}$ variant showed strong accumulation of radioactively labelled galactose, along with increased levels of arabinose and mannose, possibly due to more efficient labelling or increased production of lipoarabinomannan under restricted growth conditions (Fig. 2j, Supplementary Fig. 8b). Intriguingly, the presence of radioactive galactose was also observed in the strain complemented with the $F13A^{Wzt}$ mutant (Fig. 2j, Supplementary Fig. 8b), suggesting a functional role for this residue in the transport mechanism. Hence, this third assay appears suitable for investigating transporter variants diminishing but not stopping LLG transport.

## The cytosolic gate helix is essential for function

Once the assays were established, we complemented the *Msmeg* CRISPRi-*wzt* strain with Wzm-Wzt$_{Mabs}$ mutant in which the entire gate helix (residues $A18^{Wzt}$-$S40^{Wzt}$) was deleted and replaced with one glycine and one serine (ΔGH$^{Wzt}$). Complementation with this variant resulted in a complete growth defect akin to the catalytically-dead $E178Q^{Wzt}$ variant (Fig. 2h, Supplementary Fig. 7b, c). In addition, we observed a strong accumulation of TDM, TMM and [14C]-galactose, which are hallmarks of a non-functional Wzm-Wzt$_{Mabs}$ transporter

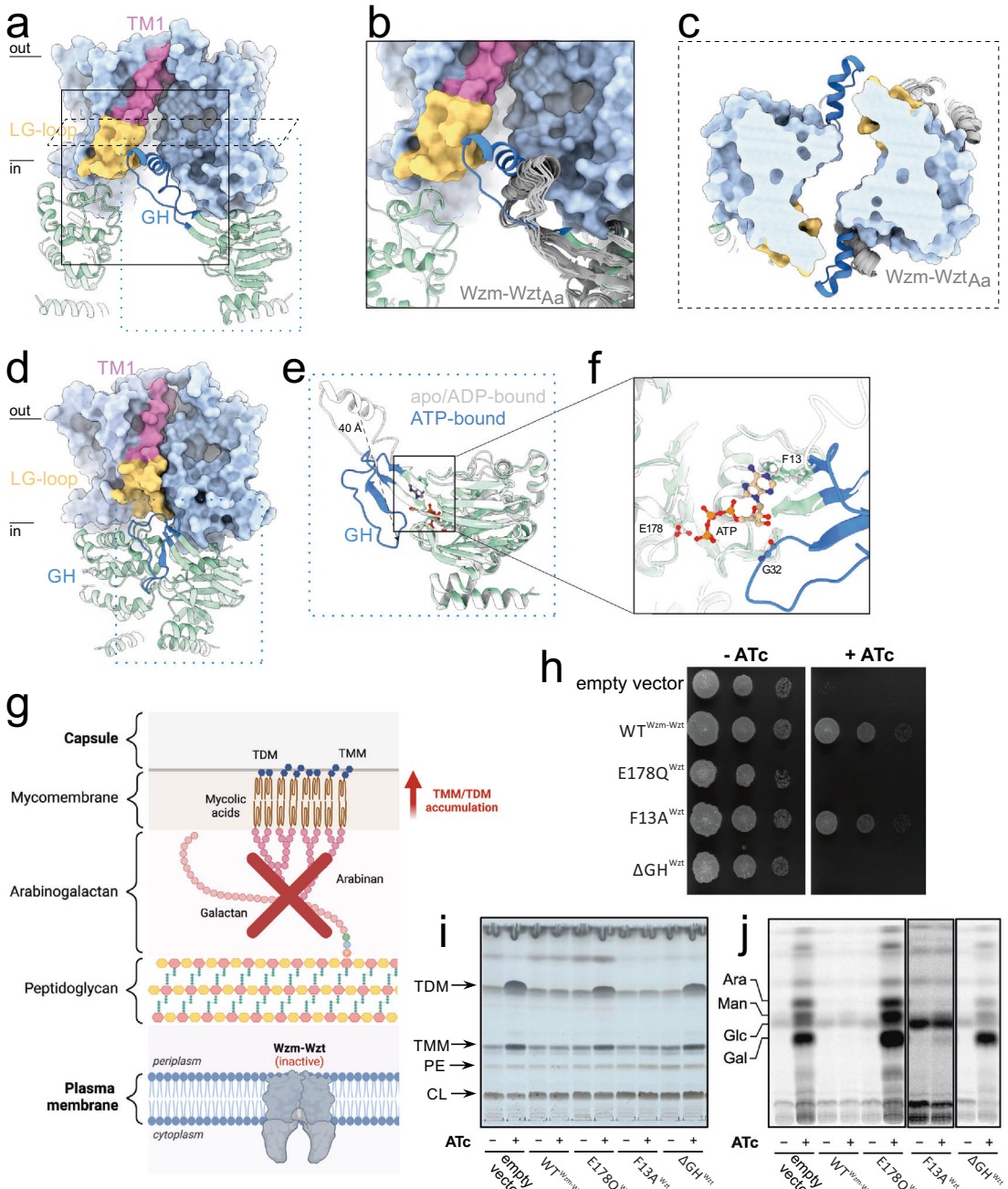

**Fig. 2 | Structure and function of the cytosolic gate helix. a** Model of apo Wzm-Wzt$_{Mabs}$ in LMNG with labelled structural elements represented as surface and NBDs as cartoon. The black box indicates zoomed view shown in panel **b** the dashed box is shown in panel **c** the dotted blue box is shown in panel **e**. **b**, **c** Superposition of Wzm-Wzt$_{Mabs}$ NBD with Wzm-Wzt$_{Aa}$ NBDs (multiple structures, grey cartoon). **b** view along the membrane plane, highlighting an alternate orientation of GH in Wzm-Wzt$_{Mabs}$. **c** view from periplasm. **d** Model of ATP-bound Wzm-Wzt$_{Mabs}$ in LMNG with labelled structural elements represented as surface and NBDs as cartoon. The dotted blue box is shown in panel **e**. **e** Superposition of Wzm-Wzt$_{Mabs}$ NBDs in the apo/ADP-bound (white cartoon) and ATP-bound (green cartoon with blue GH) conformations. The Cα-position of A31$^{Wzt}$ in GH undergoes a 40 Å movement (dotted line) during the conformational transition. Black box indicates zoomed view in panel **f**. **f** ATP binding (sticks) and its coordination by F13$^{Wzt}$. **g** Scheme of mycobacterial cell wall organization. A characteristic phenotypic feature of *wzm-wzt* repression is the overproduction of extractable cell wall lipids, trehalose monomycolates (TMM) and trehalose dimycolates (TDM), which

accumulate due to the lack of arabinogalactan that covalently binds mycolic acids. Panel g was created in BioRender. Huszár, S. (2026) https://BioRender.com/kr5o9zy. **h**–**j** Functional analysis of *Msmeg CRISPRi-wzt* strains complemented with the empty vector, wild-type Wzm-Wzt$_{Mabs}$ or NBD variants of Wzm-Wzt$_{Mabs}$ (E178Q$^{Wzt}$, F13A$^{Wzt}$, ΔGH$^{Wzt}$). **h** Evaluation of ATc-induced growth inhibition by spotting a ten-fold serial dilution of cells on agar plates. Representative images for one of the two tested clones are shown. **i** TLC analysis of lipids extracted from cells grown in the absence or presence of ATc, visualized with cupric sulfate. A representative image of three biological replicates is shown. TDM−trehalose dimycolates, TMM−trehalose monomycolates, PE−phosphatidyl ethanolamine, and CL−cardiolipin. **j** Monosaccharide composition of E-soak extracts of cells grown in radiolabeled medium in the absence or presence of ATc. TLC plates were read out by autoradiography. The F13A$^{Wzt}$ sample was separated on a different TLC plate and exposed for longer to increase the signal-to-noise ratio. Representative images of three biological replicates are shown. Ara−arabinose, Man−mannose, Glc−glucose, Gal−galactose.

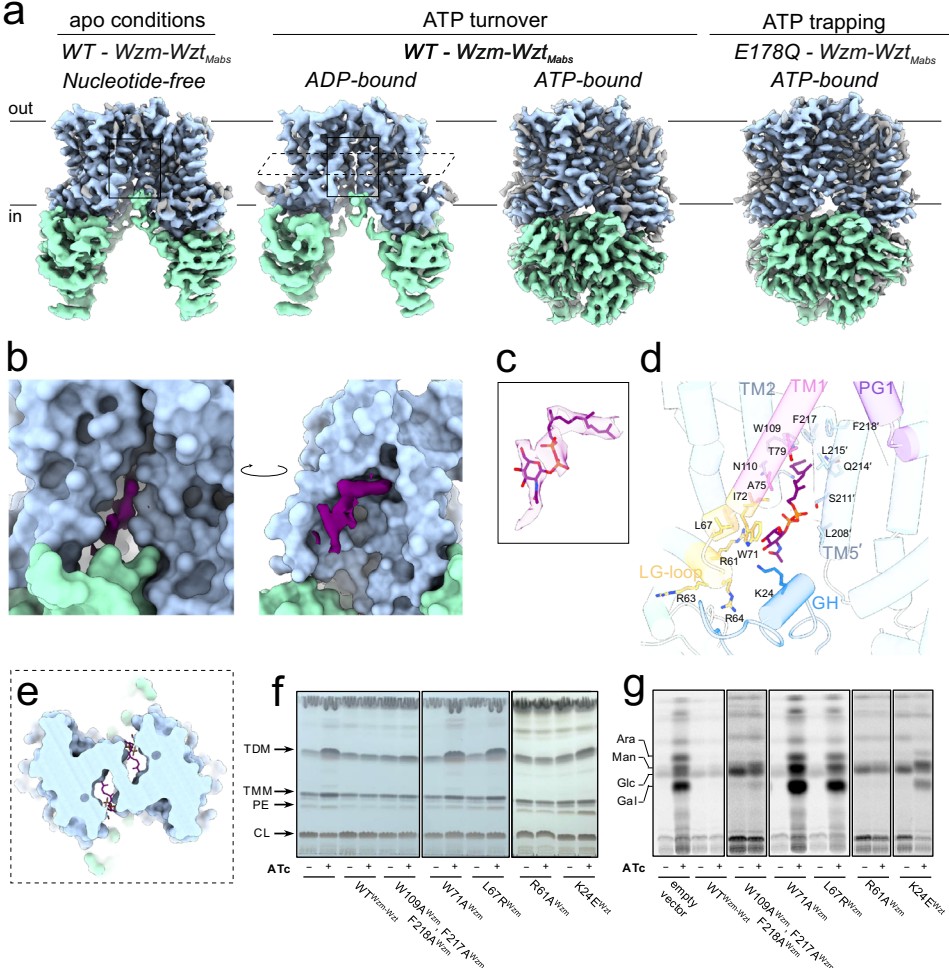

**Fig. 3 | Wzm-Wzt$_{Mabs}$ structures in nanodiscs and substrate analog binding.**
**a** Cryo-EM maps of Wzm-Wzt$_{Mabs}$ in nanodiscs. The membrane boundary is indicated and the locations of the substrate analog binding site is highlighted by a black rectangle. **b** Surface representation of the substrate analog binding cavity with non-proteinaceous cryo-EM density contoured at 6σ. **c** Farnesyl-*P-P*-Glc*p*NAc part of the substrate analog (sticks) modeled into the cryo-EM density (purple, contoured at 6σ). **d** Structural context of farnesyl-*P-P*-GlcNAc binding site (purple sticks) with contacting side chains represented as sticks. **e** Slice through the TMDs viewed from the periplasm highlighting insertion of two symmetrical substrate analog molecules. **f** TLC analysis of lipids extracted from *Msmeg CRISPRi-wzt* strains complemented with wild-type or Wzm-Wzt$_{Mabs}$ variants (W109A$^{Wzm}$-

F217A$^{Wzm}$-F218A$^{Wzm}$, W71A$^{Wzm}$, L67R$^{Wzm}$, R61A$^{Wzm}$, K24E$^{Wzt}$) grown in the absence or presence of ATc. A representative image of at least two biological replicates is shown. TDM−trehalose dimycolates, TMM−trehalose monomycolates, PE−phosphatidyl ethanolamine, and CL−cardiolipin. **g** Monosaccharide composition of E-soak extracts of cells grown in radiolabeled medium in the absence or presence of ATc. TLC plates were read out by autoradiography. The W109A$^{Wzm}$-F217A$^{Wzm}$-F218A$^{Wzm}$ variant, as well as R61A$^{Wzm}$ and K24E$^{Wzt}$ variants were separated on different TLC plates and exposed for longer to increase the signal-to-noise ratio. Representative images of at least two biological replicates are shown. Ara−arabinose, Man−mannose, Glc−glucose, Gal−galactose.

(Fig. 2i, j, Supplementary Fig. 8). To ensure that the transporter with the ΔGH$^{Wzt}$ variant was properly folded, we successfully purified it upon expression in *E. coli*, thereby excluding misfolding or aggregation as the reason for lack of function (Supplementary Fig. 6h and 9). Our results therefore demonstrate the essential role of the GH in LLG transport, thereby demonstrating that the transport mechanism of Wzm-Wzt$_{Mabs}$ differs from that of KpsMT, which naturally lacks the gate helix[24].

**Substrate analog-bound structure of Wzm-Wzt$_{Mabs}$ in nanodiscs**
We reasoned that reconstituting Wzm-Wzt$_{Mabs}$ into lipid nanodiscs might prevent the occupation of the entry channel with LMNG and thereby allow binding of the LLG substrate analog. Therefore, we purified wild-type Wzm-Wzt$_{Mabs}$ in β-DDM, reconstituted it in MSP1D1E3 nanodiscs and added farnesyl-*P-P*-Glc*p*NAc-Rha*p*-Gal*f*-Gal*f* together with ATP-Mg before preparing cryo-EM grids. For this turnover dataset, we determined two structures, corresponding to the ADP-bound conformation with disengaged NBDs and the ATP-Mg-

bound structure with closed NBDs (Fig. 3a, Supplementary Fig. 10 and 11, Supplementary Table 1). In the 4.0 Å resolution cryo-EM map of ADP-bound Wzm-Wzt$_{Mabs}$, we observed an unassigned elongated density in the groove between TM1 and TM5′ of the opposite subunit (Fig. 3b). Importantly, we also prepared a control sample without adding the substrate analog (see section below and Fig. 4). Cryo-EM analysis of the control sample revealed a different unassigned density in a similar but distinct location, which we interpreted as a phospholipid. This suggests that the density observed in the substrate analog-added sample most likely corresponds to farnesyl-*P-P*-Glc*p*NAc. Therefore, we modeled this substrate analog into the density with the farnesyl moiety entering first (termed "lipid-first" binding pose) as one possible interpretation (Fig. 3c). However, we acknowledge that the moderate resolution and incomplete density coverage leave room for alternative interpretations. The rest of the substrate likely extends outside of the transporter, making it less coordinated, more flexible, and therefore less resolved in our cryo-EM map. Farnesyl-*P-P*-Glc*p*NAc is wedged between TM1 and TM5′ of the opposite protomer and

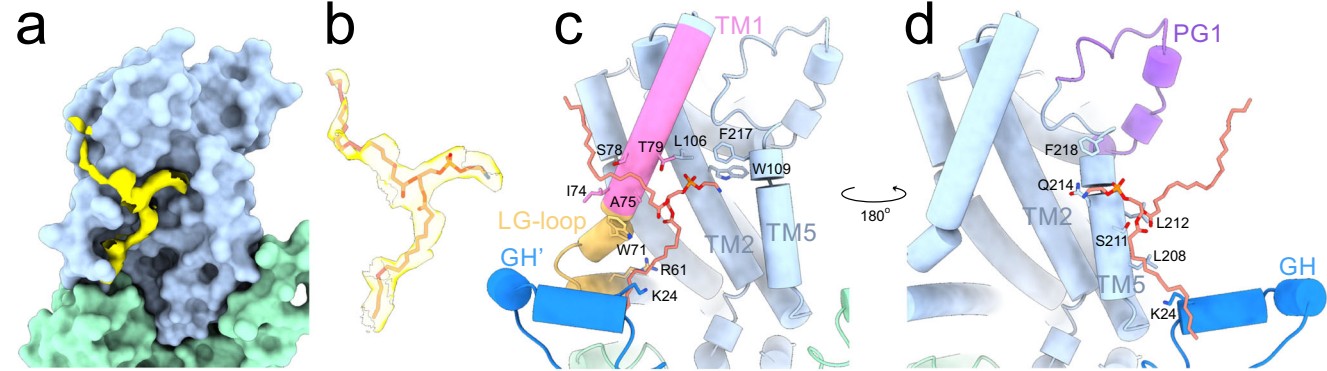

**Fig. 4 | PE lipid binding in Wzm-Wzt_{Mabs} in nanodiscs. a** Surface representation of a single Wzm-Wzt_{Mabs} protomer (apo structure in nanodiscs) featuring non-proteinaceous cryo-EM density (yellow, contoured at 4.7σ) corresponding to a phospholipid. **b** A PE lipid (sticks) modeled into the cryo-EM density (yellow, contoured at 4.7σ). Structural context of the PE lipid binding site (orange sticks) with contacting residues represented as sticks shown for the first (**c**) and the second (**d**) Wzm-Wzt_{Mabs} protomer.

interacts with residues of the central aromatic belt, the LG-loop as well as the GH that is part of the NBD (Fig. 3d, e).

The observed density could represent the substrate analog in an alternative "sugar-first" orientation wherein the Gal*f*-Gal*f*-Rha*p*-Glc*p*NAc-*P-P*-moiety enters first (Supplementary Fig. 12). However, we consider the "lipid first" binding pose of the substrate analog to be more likely for the following reasons: (i) the prenyl moiety fits better into the narrow density extending into the TMD than the sugar-chain; (ii) there is stronger density for the diphosphate moiety in the "lipid-first" binding pose versus the "sugar-first" binding pose; (iii) the prenyl chain is accommodated well in the apolar cavity by interacting with W109^{Wzm}, F217^{Wzm} and F218^{Wzm}. In further support of the "lipid-first" pose, it has been shown that the extension of the galactan chain mediated by the GlfT2 polymerase and substrate translocation are very likely coupled processes[8], which precludes the possibility that the sugar chain enters first. Finally, an analogous "lipid-first" translocation mechanism has been described for KpsMT[24]. Hence, from a physiological point of view, it is reasonable to propose that the prenyl-moiety enters first. Nevertheless, we cannot exclude the possibility that the substrate analog entered in the alternative, "sugars-first" orientation (Supplementary Fig. 12), which would then represent a non-physiological binding mode.

We also determined a cryo-EM structure of the Wzm-Wzt_{Mabs} E178Q mutant in nanodiscs, which is identical to the ATP-bound conformation of the wild-type transporter in nanodiscs in the ATP turnover dataset (RMSD 0.539 Å) (Fig. 3a, Supplementary Fig. 13, Supplementary Table 1).

### Glycolipid binding site in apo Wzm-Wzt_{Mabs}

As a control, we prepared a Wzm-Wzt_{Mabs} nanodisc sample in apo conditions without adding farnesyl-*P-P*-Glc*p*NAc-Rha*p*-Gal*f*-Gal*f* (Fig. 4, Supplementary Fig. 14, Supplementary Table 1). In the resulting cryo-EM map, we observed an unassigned density at a similar location where farnesyl-*P-P*-Glc*p*NAc-Rha*p*-Gal*f*-Gal*f* was bound to (Fig. 4a). However, the position and shape of this density are clearly different from those observed in the sample with the substrate analog. The shape resembles a lipid and hence we built the abundant *E. coli* lipid phosphatidylethanolamine (PE) into the density as one of the possible interpretations (Fig. 4b). The head group of the PE is surrounded by the central aromatic gate residues W109^{Wzm}, F217^{Wzm} and F218^{Wzm} (Fig. 4c, d). One of the hydrophobic lipid tails extends out of the TMD, while the second tail follows the cavity along TM1 and TM5 until it reaches the LG-loop and GH. The phosphate group is located 6.3–7.5 Å higher in the channel than the diphosphate of farnesyl-*P-P*-Glc*p*NAc-Rha*p*-Gal*f*-Gal*f* and is surrounded by the hydroxyl groups of S211^{Wzm} in TM5 and of

T79^{Wzm} in TM1 (Fig. 4c, d). This position of the lipid could mimic a binding pose that LLG would assume while it is threaded through Wzm-Wzt.

In summary, our cryo-EM structures revealed that the chemistry and geometry of the TMD cavity can accommodate different amphipathic molecules: the detergent LMNG, a phospholipid, and a density we interpreted as the LLG substrate analog farnesyl-*P-P*-Glc*p*NAc-Rha*p*-Gal*f*-Gal*f* (Supplementary Fig. 15).

### Functional analysis of residues interacting with LLG

Next, we used our functional assays to investigate the farnesyl-*P-P*-Glc*p*NAc-Rha*p*-Gal*f*-Gal*f* binding site by site-directed mutagenesis. In the "lipid-first" binding pose, the farnesyl part points into the cavity and is surrounded by the residues constituting the second aromatic gate, W109^{Wzm}, F217^{Wzm} and F218^{Wzm} (Fig. 3d). In KpsMT, the analogous residues, F85, Y194 and F195 are in close proximity to the phosphatidylglycerol lipid tails[24] (Supplementary Fig. 16). To explore the importance of this aromatic environment, we generated a triple variant by substituting these amino acids with alanine and tested them in our CRISPRi strain. These mutations did not appear to affect growth or TMM and TDM production, but they resulted in accumulation of radiolabeled galactose, similar to the A-loop variant (F13A^{Wzm}) used in our test assays (Fig. 3f, g, Supplementary Fig. 7b, c and Supplementary Fig. 8). Hence, the transport activity of this triple variant was only slightly diminished.

The position of the conserved W71^{Wzm} residue in the LG loop is also conspicuous. The indole group of W71^{Wzm} forms stacking interactions with the Glc*p*NAc-moiety of LLG analog (Fig. 3d). When we mutated W71^{Wzm} to alanine, the result was a lethal phenotype, confirming the crucial function of this residue in the transporter's operation (Fig. 3f, g, Supplementary Fig. 6c, 7b, c and 8). To further investigate the role of the LG-loop in gating, we mutated its second conserved residue, L67^{Wzm}. Complementation of the CRISPRi strain with the L67R^{Wzm} variant also resulted in a lethal phenotype (Fig. 3f, g, Supplementary Fig. 6c, 7b, c and 8). All variants of Wzm-Wzt_{Mabs} were purified and confirmed to be properly folded by size exclusion chromatography and expressed in complemented CRISPRi strains (Supplementary Fig. 6h and 7d).

It is interesting to note that, in contrast to KpsMT, Wzm-Wzt_{Mabs} lacks positively charged residues in the TMD channel that could coordinate the diphosphate group. In KpsMT, residues R89 and R94, which were shown to be essential for CPS transport, along with R187, which coordinates the phosphate in the cryo-EM structure of glycolipid 2-bound KpsMT[24], are not conserved in Wzm-Wzt_{Mabs}. Instead, N113, E118 and G210 are located at the respective positions

(Supplementary Fig. 16). The only exception is an arginine residue in the LG-loop, present in both KpsMT (R35) and in Wzm-Wzt$_{Mabs}$ (R61$^{Wzm}$), which was shown to be critical for KpsMT functioning[24]. We therefore mutated R61$^{Wzm}$ in Wzm-Wzt$_{Mabs}$ to alanine, but did not observe any effect on LLG transport in our functional assays (Fig. 3f, g, Supplementary Figs. 7b; 8a, c), highlighting differences with KpsMT and suggesting that diphosphate coordination in Wzm-Wzt$_{Mabs}$ may rely on different residues. An interesting candidate for diphosphate coordination is K24$^{Wzt}$, which is part of the GH. In the ADP-bound conformation, K24$^{Wzt}$ forms a hydrogen bond with GlcNAc of the LLG substrate and is well-positioned to interact with the negatively charged diphosphate moiety (Fig. 3d and Supplementary Fig. 12e). Mutation of this K24$^{Wzt}$ to glutamate showed a clear growth defect in liquid culture, accumulation of TDM, TMM and of radioactively labelled galactose (Fig. 3f, g, Supplementary Figs. 6h, 7b; 8a, c), which are hallmarks of a functionally compromised Wzm-Wzt$_{Mabs}$ transporter.

### Conformational differences in LMNG versus nanodiscs

In the ATP-Mg-bound structure from the turnover dataset in nanodiscs, we observed that the position of the NBDs is different compared to the corresponding structure determined in LMNG (Supplementary Fig. 17a, b). By measuring the opening of the NBDs as the distance between the C$_\alpha$-positions of Walker A K71$^{Wzt}$ and ABC signature S154$^{Wzt}$, we noted that the ATP-Mg-bound structure in LMNG features NBDs that are not fully locked (10 Å distance compared to 9.5 Å in nanodiscs). The structural difference is most pronounced at the C-terminus of NBDs: in LMNG, the NBDs of the ATP-bound transporter are 4.5 Å further apart than in the nanodiscs (measured between C$_\alpha$-positions of A253$^{Wzt}$ at the C-terminal NBD helix) (Supplementary Fig. 17a, b).

We further noted that in the structure of ATP-Mg-bound Wzm-Wzt$_{Mabs}$ determined in nanodiscs, the relative orientation of the two TMD subunits is twisted by a 10° rotation compared to respective structure determined in LMNG, making the TMD-dimer more compact in nanodiscs (Supplementary Fig. 17c–f). The interface surface area between the two TMD subunits, as calculated by PDBePISA, increases from 1361 Å$^2$ in LMNG to 1688 Å$^2$ in nanodiscs (Supplementary Fig. 17e). Intriguingly, six hydrogen bonds and a salt bridge stabilize the TMD dimer in the structure determined in LMNG (Supplementary Fig. 17f). In contrast, these interactions are disrupted in the nanodisc environment, thereby weakening the contact between TM1 and TM5′ of the opposite subunit at the cytosolic end (Supplementary Fig. 17f). Due to these conformational rearrangements, the substrate-binding cavity between TM1 and TM5′ changes its shape from a relatively flat surface in LMNG to a convex shape in nanodiscs (Supplementary Fig. 17c, d). Additionally, the periplasmic cavity between TM1 and PG1′, through which the glycolipid is expected to pass, is smaller in nanodiscs. In more general terms, our structures determined in LMNG versus nanodiscs suggest high conformational freedom of the TMDs to modulate the shape of the enclosed cavity and the interactions at the periplasmic exit gate in response to lipid and LMNG interactions (Supplementary Fig. 16). This conformational flexibility, which is also reflected in the large conformational differences observed for Wzm-Wzt$_{Aa}$ structures determined in detergent and nanodiscs (Supplementary Figs. 18 and 19), appears to be a molecular hallmark of polysaccharide ABC transporters.

Regardless of whether we determined our structures in LMNG or nanodiscs, all ATP-bound cryo-EM structures of Wzm-Wzt$_{Mabs}$ feature a closed periplasmic exit and a discontinuous channel (Fig. 1b and Supplementary Fig. 18). This contrasts with previous work on Wzm-Wzt$_{Aa}$, for which a continuous TMD channel was reported in two X-ray structures (6OIH[18] and 6M96[19]) and one cryo-EM structure (8DLO[21]) (Supplementary Fig. 18). Because an open channel is very unlikely to exist under physiological conditions, we consider our structures to represent more physiologically realistic conformations. Opening of

the channel is expected to be a transient event that is initiated upon substrate binding and translocation. In Wzm-Wzt$_{Mabs}$, the exit gate is blocked by the Y87$^{Wzm}$ residue located on TM1, which is part of the periplasmic aromatic belt and likely needs to rearrange to allow LLG translocation (Supplementary Fig. 6d). We therefore mutated Y87$^{Wzm}$ to alanine and observed an intermediate phenotype after complementing the CRISPRi strain (Supplementary Fig. 7b, c and 8). These findings suggest that while Y87$^{Wzm}$ is involved in LLG transport, it is not essential for the process.

## Discussion

Polyprenyl-phosphate lipids are often critical for translocation of polar molecules, such as polysaccharides, across hydrophobic membranes, a process essential for the biosynthesis of complex cell wall components in various organisms. However, the underlying reasons for lipid attachment to polysaccharides and the mechanisms driving their translocation remain enigmatic. A plausible reason for attaching sugar chains onto branched polyprenyl-phosphate carriers might be associated with the biophysical challenge of translocating polar and bulky polysaccharides across the membrane. The lipidic moiety not only anchors the polysaccharide to the membrane and thereby increases local concentration, but it may also serve as a handle recognized by transport proteins to initiate the translocation reaction. Our LLG analog bound structure wherein the polyprenyl-moiety is suggested to enter the transporter first supports this hypothesis, as well as the lipid-bound structures of KpsMT showing insertion of the lipid tails into the transporter[24]. The branched nature of the polyprenyl-moiety may further support specific recognition by the transporter, because the great majority of bacterial lipids consist of unbranched fatty acids.

Intriguingly, LLG features the full spectrum of biophysical properties, namely the apolar decaprenyl moiety containing 50 carbon atoms, followed by two negatively charged phosphate groups and finally a chain of around 30 polar sugar units, which are mostly galactofuranose residues. Its size is very large relative to the dimensions of the lipid bilayer and the Wzm-Wzt transporter sitting therein. Hence, in order to catalyze substrate transport, the Wzm-Wzt transporter requires astonishing capabilities: i) it must specifically recognize LLG and initiate its translocation, ii) it must translocate LLG in a stepwise manner owing to its size and iii) it must deal with the "contradicting" physical properties of LLG during the translocation, including the negatively charged diphosphate moiety.

Based on our own work and that of others[24], we propose the following translocation mechanism (Fig. 5). Polymerization of galactan is initiated by the GlfT2 enzyme, which extends its substrate, dec-aprenyl-P-P-Glc$p$NAc-Rha$p$-Gal$f$-Gal$f$ (Step 1). Because in cells lacking Wzm-Wzt the chain length of the LLG increases considerably[8], it is plausible to assume that GlfT2 interacts with Wzm-Wzt to coordinate LLG synthesis and export. LLG is suggested to enter the cavity of ADP-bound Wzm-Wzt having the NBDs separated (Step 2). The lipid tail (or at least parts of it) is proposed to interact with the central aromatic belt, as suggested by the "lipid-first" binding pose in the LLGanalog-bound structure. The GH loop undergoes a dramatic structural rearrangement upon binding of ATP, which may assist loading of the substrate to the transporter (Step 3). In a process for which we lack structural evidence for any polysaccharide ABC transporter, the sugar-chain of LLG is most likely inserted into a continuous channel formed at the TMD interface, which is coated with the three aromatic belts (Steps 4 and 5). Both GH and the LG-loop interact with the LLG analog and are suggested to assist the substrate movement, because they are essential for transport. The phospholipid binding site of the ADP-bound structure of Wzm-Wzt$_{Mabs}$ in nanodiscs (Fig. 4) might mark the path along which the decaprenyl-moiety could move from its initial binding site as visualized in the LLG analog bound structure (Fig. 3) towards the hydrophobic core of the lipid bilayer. A closer inspection

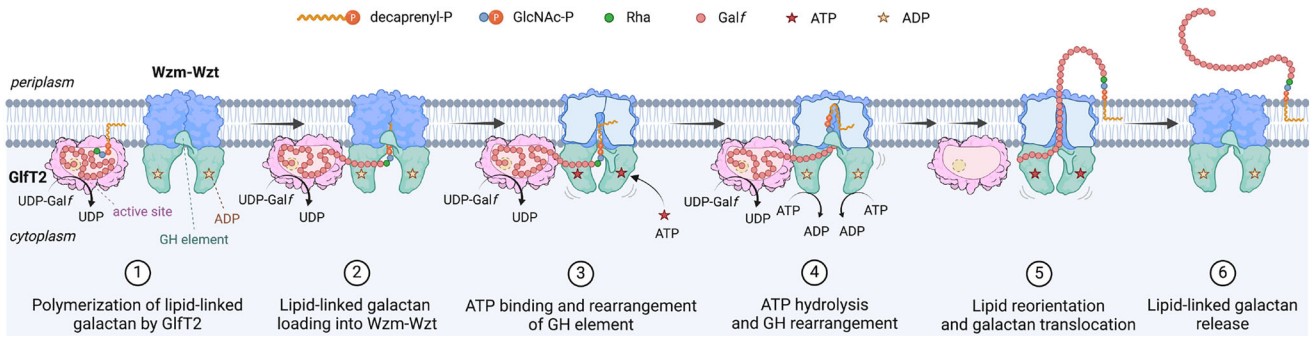

**Fig. 5 | Proposed LLGtransport mechanism.** Details are provided in the main text. This figure was created in BioRender. Huszár, S. (2026) https://BioRender.com/oim5sh4.

of the phospholipid binding site (Fig. 4) reveals that one of the lipid tails sticks out of the cavity, and thereby crosses TM1 and TM5′ at the dimer interface. We therefore envisage that the decaprenyl-moiety would as well first cross TM1 and TM5′ at the dimer interface to become fully embedded in the lipid bilayer, followed by the translocation of the diphosphate moiety dragging along the polysaccharide chain within the transporter. The architecture of type V ABC transporters allows for such a mechanism; the substrate would slide along the TM1-TM5′ interface and slightly displace the two helices on the way up towards the periplasmic exit gate. Thereby, LLG would finally reach the configuration as shown in step 5, in which the decaprenyl-moiety is attaching LLG at the outer leaflet of the membrane, the diphosphate-moiety is exposed to the periplasm and the polysaccharide is inserted in a continuous channel within Wzm-Wzt coated with aromatic residues, and is ready to be threaded through the channel in a processive manner at the expense of ATP hydrolysis. Extensive aromatic interactions with translocating sugar chains are a common theme in polysaccharide transport machineries[29]. Their likely role is to facilitate sugar chain sliding through the channel by hydrophobic interactions[30–32], as well as to seal the channel in order to prevent the undesired flux of solutes such as ions or nutrients during the translocation process.

Our work provides potential clues about energy transduction. Conformational changes invoked by ATP binding and hydrolysis result in NBD closure and disengagement. These changes are coupled to large movements of the GH element within the NBDs, which are proposed to support LLG loading into the transporter as well as they may facilitate oligosaccharide movement during each ATP hydrolysis cycle. In addition, the inside-negative/outside-positive membrane potential could energetically contribute to the translocation of the diphosphate moiety.

In conclusion, our work presents a set of highly resolved structures of a mycobacterial polysaccharide ABC transporter in different conformations, and provides molecular insights into its interaction with a natural ligand analog. Owing to its essential nature, arabinogalactan biosynthesis represents an attractive target for antibiotic development, as is evidenced by the fact that the important first-line tuberculosis drug ethambutol targets mycobacterial arabinosyltransferases, which act downstream of Wzm-Wzt to attach arabinose moieties to the translocated LLG. Moreover, *wzm-wzt* genes proved to be highly vulnerable in *M. tuberculosis*[33], making them high-value targets for development of urgently needed new drugs for tuberculosis and emerging infections caused by nontuberculous mycobacteria. The structures of Wzm-Wzt*Mabs* presented in this work provide a rationale for the targeted development of drugs that either block the LLG binding site and/or interfere with the movement and LLGinteractions of the GH and LG loop, two structural elements which we demonstrate here to be of key importance for LLG transport and thus cell wall biogenesis.

## Methods

### Cloning, expression and purification of Wzm-Wzt*Mabs*

The *wzt*Mabs and *wzm*Mabs genes were amplified from the pINIT construct carrying the *MAB_0203c-MAB_0202c-MAB_0201c* operon of *Mycobacterium abscessus* ATCC_19977 using Q5 DNA polymerase (NEB). For the amplification of *wzm*, the primers 5′ − CAT CAC CAC AGC CAG ATG AGC AGT TCG GCC GGC CTG and 5′ − GGC GCG CCG AGC TCG CTA AAC CCA ATA CGG GAC CCT GG were used, while for *wzt*, the primers 5′ − AGA AGG AGA TAT ACA ATG TCA GAT ATC CAA ACT CAC CA and 5′ − CAG ACT CGA GGG TAC CTA GCC ATC GTG CGC GTC TT were used. The amplified fragments were cloned into pETDuet-1 vector (Novagen) with an N-terminal histidine tag on Wzm using GenBuilder DNA Assembly (GenScript).

Terrific Broth medium supplemented with ampicillin (100 µg/ml) was inoculated 1:50 with freshly transformed *E. coli* C43(DE3) strains. The cells were grown at 37 °C while shaking until an $OD_{600}$ of ∼0.5 was reached. The temperature was then lowered to 25 °C and after 1 h, the cultures were induced with 0.5 mM isopropyl β-D-thiogalactoside and incubated overnight while shaking. The cells were harvested by centrifugation at 6500 × *g* for 15 min at 4 °C and resuspended in approximately 1/36 of the original culture volume in the lysis buffer (20 mM Tris-HCl pH 8, 200 mM NaCl, 3 mM $MgCl_2$, a spatula tip of DNase). The cell suspension was lysed using an M-110P microfluidizer (Microfluidics) with three passes at 25 kPa. The lysate was centrifuged for 15 min at 10,000 × *g* and 4 °C to remove cell debris, and then the supernatant was centrifuged for 2 h at 170,000 × *g* and 4 °C. The membrane vesicle fraction contained in the pellet was homogenized in approximately 1/3 of lysate volume in TBS with 10 % glycerol, flash-frozen with liquid nitrogen, and stored at −80 °C until further use.

Membrane vesicles were solubilized with 1% (w/v) n-dodecyl-β-D-maltoside (β-DDM, Glycon Biochemicals) or lauryl-maltose-neopentyl-glycol (LMNG, Anatrace) detergent for 1 h on a rotator at 4 °C. The extract was then centrifuged again at 170,000 × *g* for 30 min at 4 °C to remove insoluble material. Imidazole solution (pH 8.0) was added to the supernatant to a final concentration of 30 mM, which was mixed with Ni-NTA Superflow resin (Qiagen). Protein binding was carried out for 1 h at 4 °C on a rotator, after which the mixture was poured onto a gravity flow column. The Ni-NTA column was washed with 20 column volumes of wash buffer (20 mM Tris-HCl pH 8, 300 mM NaCl, 30 mM imidazole pH 8, 0.03% β-DDM or 0.002% LMNG), and His-tagged proteins were finally eluted with four column volumes of elution buffer (20 mM Tris-HCl pH 8, 300 mM NaCl, 300 mM imidazole pH 8, 0.03% β-DDM or 0.002% LMNG). Fractions with the highest absorption at 280 nm and the lowest 260 nm/280 nm ratio were concentrated using an Amicon Ultra Centrifugal Filter 30 kDa MWCO (Merck Millipore). The proteins were further purified at 4 °C by size-exclusion chromatography on an Äkta Pure (Cytiva) machine using a Superdex 200 Increase 10/300 GL size-exclusion column (Cytiva) in SEC buffer (20 mM Tris-HCl pH 8, 300 mM NaCl, 0.03% β-DDM or 0.002% LMNG).

Fractions exhibiting the size range expected for correctly assembled Wzm-Wzt$_{Mabs}$ were analyzed by SDS-PAGE and further concentrated if necessary. Purified proteins were used immediately or flash-frozen in liquid nitrogen and stored at −80 °C.

### Reconstitution of Wzm-Wzt$_{Mabs}$ in nanodiscs

Purified Wzm-Wzt$_{Mabs}$ E178Q$^{Wzt}$ variant solubilized in β-DDM was reconstituted into MSP1D1E3 nanodiscs (prepared in house) in a molar ratio of 100:5:1 (lipids:MSP1D1E3:Wzm-Wzt$_{Mabs}$). First, the mixture of protein and *E. coli* polar lipids plus phosphatidylcholine (3:1, w/w, Avanti) was incubated at 4 °C for 30 min with end-over-end rotation before adding MSP1D1E3 protein and incubating again at 4 °C for 30 min. Then, 120 mg BioBeads (BioRad) per 600 μl of reaction mixture were added and the mixture was incubated at 4 °C overnight with end-over-end rotation. BioBeads were removed by filtering and empty nanodiscs were removed by gel-filtration using a Superdex 200 Increase 30/100 GL column (Cytiva) equilibrated with TBS buffer.

For preparing the wild-type transporter in nanodiscs, Wzm-Wzt$_{Mabs}$ was purified in β-DDM in the presence of 4 mM ATP and 10 mM MgCl$_2$ in all purification buffers to increase protein stability. Reconstitution was performed as described above for the E178Q$^{Wzt}$ variant using His-tag-cleaved MSP1D1E3. To remove empty nanodiscs, the reconstituted mixture was incubated with Ni-NTA Superflow resin (Qiagen) for 1 h at 4 °C on a rotator and Wzm-Wzt$_{Mabs}$-containing nanodiscs were eluted with TBS supplemented with 300 mM imidazole pH 8 and further purified by size exclusion chromatography using a Superdex 200 Increase 30/100 GL column (Cytiva) in TBS buffer. Purified nanodiscs were immediately used for preparing cryo-EM grids.

### Cryo-EM sample preparation and data collection

Freshly purified Wzm-Wzt$_{Mabs}$ in LMNG and Wzm-Wzt$_{Mabs}$ in nanodiscs were concentrated to 2 mg/ml using an Amicon Ultra-0.5 ml concentrating device (Merck) with a 100 kDa filter cutoff. For the apo conditions (datasets #1 and #4) the protein was applied on the grids right away. For the ATP turnover and ATP trapping conditions (datasets #2, #3, #5 and #6) a small aliquot of Wzm-Wzt$_{Mabs}$ was mixed with 50 μM farnesyl-*P-P*-Glc*p*NAc-Rha*p*-Gal*f*-Gal*f* (synthesized as reported earlier[25]), 2.5 mM ATP-Mg$^{2+}$ and incubated for 30 s at 25 °C. 3 μl of the sample was applied onto the holey-carbon cryo-EM grids (Au R1.2/1.3, 300 mesh, Quantifoil), which were prior glow discharged at 15 mA for 60 s (Pelco easiGlow), blotted for 4–5 s and plunge frozen into a liquid ethane/propane mixture with a Vitrobot Mark IV (Thermo Fisher) at 4 °C and 100% humidity. Samples were stored in liquid nitrogen until further use.

Four datasets (wild-type in LMNG in apo (dataset #1) and ATP turnover conditions (dataset #2), wild-type in nanodiscs in ATP turnover conditions (dataset #5), E178Q in nanodiscs in ATP trapping conditions (dataset #6)) were collected on a 300 keV Titan Krios G3i microscope (Thermo Fisher) with a post-column BioQuantum energy filter with a 20 eV slit and a 100 μm objective aperture. Images were acquired in an automatic manner with EPU v2.9 on a K3 direct electron detector in super-resolution mode with two-times binning at a nominal magnification of 130,000x (0.65 Å physical pixel size for binned data) and a defocus range from −0.8 to −2.2 μm. For Wzm-Wzt$_{Mabs}$ in LMNG in apo conditions (dataset #1) the total exposure dose was 63.36 electrons/Å2, 1.3 s exposure time, 41 frames. For Wzm-Wzt$_{Mabs}$ in LMNG in ATP turnover conditions (dataset #2): 64.30 electrons/Å$^2$ total exposure dose, 1.3 s exposure time, 36 frames. For Wzm-Wzt$_{Mabs}$ in nanodiscs in ATP turnover conditions (dataset #5): 60.00 electrons/Å2 total exposure dose, 1.1 s exposure time, 36 frames. For E178Q in nanodiscs in ATP trapping conditions (dataset #6): 66.46 electrons/Å2 total exposure dose, 1.3 s exposure time, 36 frames.

Two datasets (E178Q in LMNG in ATP trapping conditions (dataset #3), wild-type in nanodiscs in apo conditions (dataset #4), were collected on a 200 keV Glacios G2 microscope (Thermo Fisher) using a 100 μm objective aperture and a 20 μm C2 aperture. Images were acquired in an automatic manner with EPU v2.9 on a Falcon 4i direct electron camera at a nominal magnification of 190,000x (0.72 Å calibrated pixel size) and a defocus range from −0.8 to −2.2 μm. For E178Q in LMNG in ATP trapping conditions (dataset #3) the total exposure dose was 60 electrons/Å2, 5.27 s exposure time, 45 frames. For Wzm-Wzt$_{Mabs}$ in nanodiscs in apo conditions (dataset #4): 60.21 electrons/Å2 total exposure dose, 6.06 s exposure time, 45 frames.

### Cryo-EM image processing

Collected data were pre-processed on-the-fly in CryoSPARC Live including beam-induced motion correction with MotionCor2 and estimation of CTF parameters using Patch CTF method. Particles were picked using either blob or templates generated from manually picked particles. Picked particles were extracted with a box size of 400 pixels and Fourier cropped to 200 or 100 pixels and imported into CryoSPARC v4.1-4.6 for all further processing steps. Extracted particles were subjected to 2D classification, followed by ab-initio reconstruction (C1 symmetry) and non-uniform refinement (C1 and C2 symmetry). The 100 pixels cropped particles were re-extracted with a box size of 400 pixels and Fourier cropped to 200 pixels before the refinement jobs.

For processing of wild-type Wzm-Wzt$_{Mabs}$ in nanodiscs (datasets #4 and #5) several rounds of heterogeneous refinements were used for better particle separation. On the final stages of processing of these two datasets, a 3D classification job with a focus mask around TMD was used for separating ligand-bound protein particles from ligand-free (see Supplementary Figs. 2, 3, 5, 9, 11, 12 for specific details on the data processing pipeline). After careful inspection of each class resulting from focused 3D classification in case of the ATP turnover dataset (datasets #5), we observed separation of substrate analog-bound particles from ligand-free particles and from what seems to be PE lipid-bound particles (Supplementary Fig. 10). For all datasets, the potential C2 symmetry application bias was checked at all stages of data processing by performing refinement jobs with C1 symmetry imposed. The directional resolution anisotropy of density maps was quantitatively evaluated using the 3DFSC web interface (https://3dfsc.salk.edu/)[34].

### Model building, refinement and validation

The models were built in Coot[35] using the AlphaFold2 prediction of Wzm-Wzt$_{Mabs}$ or determined Wzm-Wzt$_{Mabs}$ structures as reference. The resolutions of the maps were of sufficient quality to unambiguously assign the protein sequence and model Wzt residues and most of the Wzm residues excluding its first 30 amino acids. For building the gate helix, the maps were sharpened to different B-factors and low-pass filters and manually checked in Coot and ChimeraX. After visual inspections in Coot, the models were analyzed in ISOLDE[36]. Real-space refinements were performed in Phenix[37]. CC scores (correlation coefficients) are calculated in Phenix and indicate how well the atomic model agrees with the experimental cryo-EM map. CC(mask) is reported for each model in Supplementary Table 1. The Q-score is a per-atom or per-residue quality metric that reflects how well individual atoms or residues of the model fit the cryo-EM density map. It is complementary to CC scores but provides local, resolution-dependent detail. The overall mean Q-score calculated in ChimeraX for each model is reported in Supplementary Table 1. Figures were prepared using ChimeraX[38]. Cavities were calculated using 3 V web server (shell probe radius of 6 Å, solvent-excluded probe radius of 1.5 Å)[39].

### Characterization of the complemented *Msmeg* CRISPRi-*wzt* strains

The *Msmeg* PLJR962-*wzt*$_{SM}$ (*Msmeg* CRISPRi-*wzt*) strain used for functional studies was previously characterized[8]. The plasmid pFLAG_OriM carrying *wzt-glfT1-wzm*$_{Mabs}$ for complementation was prepared as

described[27] with minor changes. Briefly, the *wzt-glfT1-wzm*$_{Mabs}$ operon was amplified using Q5 DNA polymerase (NEB) from *M. abscessus* gDNA using primer pairs 5′ – ATA TAT GCT CTT CTA GTG TCA GTA TCC AAA CTC ACC AGG CGT GG and 5′ – TAT ATA GCT CTT CAT GCC TAA ACC CAA TAT GGA ACC CTG GCC CGA TA designed by the FX primer tool (https://www.fxcloning.org). A stop codon was added to the reverse primer to prevent the expression of a C-terminal 3xFLAG tag, which we expected to interfere with transporter assembly. The amplicon was then cloned into pINIT (Addgene plasmid #46858) according to the published protocol[27]. The targeted operon in pINIT-*wzt-glfT1-wzm*$_{Mabs}$ was confirmed by sequencing and subsequently sub-cloned into pFLAG_OriM (Addgene plasmid #110097). The pFLAG_OriM-*wzt-glfT1-wzm*$_{Mabs}$ construct, as well as the mutated versions of pFLAG_OriM-*wzt-glfT1-wzm*$_{Mabs}$ (prepared as described below) were then electroporated into *Msmeg* CRISPRi-*wzt*. The control strain represented *Msmeg* CRISPRi-*wzt* transformed with the empty pFLAG_OriM.

For an initial evaluation of the growth, two clones for the control and complemented *Msmeg* CRISPRi-*wzt* strains were plated in serial dilutions on solid 7H11 medium supplemented with 10% (v/v) OADC, kanamycin (20 µg/ml), hygromycin (50 µg/ml) in the absence or presence of ATc (100 ng/ml). Each plate with mutated pFLAG_OriM-*wzt-glfT1-wzm*$_{Mabs}$ contained the control strain *Msmeg* CRISPRi-*wzt* transformed with the empty pFLAG_OriM and the *Msmeg* CRISPRi-*wzt* strain transformed with the wild-type version of pFLAG_OriM-*wzt-glfT1-wzm*$_{Mabs}$. Plates were cultivated at 37 °C for 2–4 days until visible colonies formed. The growth of the two clones was comparable in each tested strain and one of them was selected for further experiments. The growth of these strains in liquid medium was analyzed in glycerol-alanine-salts (GAS) medium supplemented with tyloxapol (0.025%), kanamycin (20 µg/ml), hygromycin (50 µg/ml) in the absence or presence of ATc (100 ng/ml) at 37 °C and 120 rpm. The media were inoculated with fresh early-log pre-cultures to the initial $OD_{600} = 0.01$.

At 45 h of growth, 10 ml of *Msmeg* cultures (control and complemented *Msmeg* CRISPRi-*wzt* strains) were collected by centrifugation ($3000 \times g$, 10 min, RT) and subjected to stepwise extractions with $CHCl_3/CH_3OH$ to obtain lipids. First, the cell pellets were treated with 3 ml of $CHCl_3/CH_3OH$ (1:2) at 56 °C for 2 h, followed by 3 ml of $CHCl_3/CH_3OH$ (2:1) for 2 and 1 h, respectively. The extracts were combined, dried under $N_2$ and washed twice with a biphasic wash mixture of $CHCl_3/CH_3OH/H_2O$ (4:2:1). The organic phase was dried under $N_2$ and dissolved in $CHCl_3/CH_3OH/NH_4OH/H_2O$ (65:25:0.5:3.6) in the volume corresponding to 83 µl of solvent mixture per 10 ml culture of $OD_{600} = 1$. Aliquots of 10 µl of lipid extracts were loaded on Silica gel 60 $F_{254}$ TLC plates (Merck) and developed in a mixture of $CHCl_3/CH_3OH/H_2O$ (20:4:0.5). Lipids were visualized using 10% (w/v) $CuSO_4$ in 8% (v/v) phosphoric acid.

## Site-directed mutagenesis

Mutant versions of Wzm-Wzt$_{Mabs}$ were generated with the Quick-Change method using either pINIT-*wzt-glfT1-wzm*$_{Mabs}$ or pET-Duet-1-*wzt-wzm*$_{Mabs}$ as templates. The primers used for generating mutations are listed in Supplementary Table 2 and the presence of mutations was confirmed by sequencing. For pINIT, the operon carrying the mutation was subsequently sub-cloned into pFLAG_OriM as described above and the resulting construct was sequenced again. For W109A$^{Wzm}$, F217A$^{Wzm}$, F218A$^{Wzm}$ triple mutant, pINIT-*wzt-glfT1-wzm*$_{Mabs}$ F217A$^{Wzm}$, F218A$^{Wzm}$ or pET-Duet-1-*wzt-wzm*$_{Mabs}$ F217A$^{Wzm}$, F218A$^{Wzm}$ double mutants were used as templates for the PCR reactions using primers designed for introducing W109A$^{Wzm}$ mutation.

## Gene expression analysis by droplet-digital PCR (ddPCR)

*Msmeg* strains (control and complemented *Msmeg* CRISPRi-*wzt*) were cultivated as described above. At 24 h of growth, 10 ml of cultures were collected by centrifugation ($3000 \times g$, 10 min, 4 °C). RNA was extracted using the Quick-RNA Fungal/Bacterial Microprep Kit (Zymo Research) according to the manufacturer's instructions. Cells were disrupted by FastPrep-24 (MP Biomedicals) in four 40 s cycles at 6 m/s. 1 µg of RNA samples were treated with DNase using the TURBO DNA-free™ Kit (Invitrogen, Thermo Fischer Scientific). RNA integrity was analyzed in 1% DEPC-treated agarose gel. 200 ng of RNA was used for cDNA synthesis by iScript™ cDNA Synthesis Kit (Bio-Rad). 2 µl of 4-fold diluted (for *Msmeg* operon) or 40-fold diluted (for *Mabs* operon) cDNA sample was used in ddPCR reaction mixture using the QX200™ ddPCR™ EvaGreen Supermix (Bio-Rad) and the corresponding primer pair (150 nM each, Supplementary Table 3). Gene expression was quantified by ddPCR QX200 system (Bio-Rad) and target concentrations (copies/µl) were normalized to *sigA* transcript (housekeeping gene). Primer pairs specific for individual genes of the *Msmeg* or *Mabs* *wzt-glfT1-wzm* operon were designed using the Primer3Plus software[40].

## Whole cell radiolabeling, extraction and analysis

*Msmeg* strains (control and complemented *Msmeg* CRISPRi-*wzt*) were cultivated in GAS medium supplemented with tyloxapol (0.025%), kanamycin (20 µg/ml), hygromycin (50 µg/ml), and with or without ATc (100 ng/ml) as described above. After 45 h of the growth (mid-log phase), the 10 ml cultures were labeled with 0.5 µCi/ml [$^{14}$C] D-glucose (ARC, specific activity 300 mCi/mmol) for 3 h with shaking. The cells were collected by centrifugation ($3000 \times g$, 4 °C, 10 min) and washed twice with saline. Cell pellets were subjected to a series of extractions using 2 ml each of the following: hot 96% ethanol for 20 min at 70 °C with stirring; $CHCl_3/CH_3OH/H_2O$ (10:10:3) and "E-soak" [$H_2O$/ethanol/diethyl ether/pyridine/$NH_4OH$ (15:15:5:1:0.017)], both for 30 min at 70 °C with stirring. The monosaccharide composition in the E-soak extracts was determined in aliquots of 1.8 ml of E-soak extract per $OD_{600} = 0.2$ of culture. The extracts were dried under the stream of nitrogen, and hydrolyzed with 200 µl of 2 M trifluoroacetic acid (TFA) at 120 °C. TFA was evaporated under the stream of nitrogen and the hydrolysates were repeatedly washed by a few drops of $CH_3OH$ (2 ×) and $H_2O$ (2 ×). Two ml of $CHCl_3/H_2O$ (1:1) were added to the samples, mixed and centrifuged. Water phases were removed, dried and separated on high-performance thin-layer chromatography silica gel plates developed twice in ethyl acetate/pyridine/glacial acetic acid/$H_2O$ (6:3:1:1). The monosaccharides were detected with an α-naphthol detection reagent [1% (w/v) α-naphthol and 5% (v/v) $H_2SO_4$ in ethanol]. Radioactive signals were visualized by phosphor imaging (Amersham Typhoon 5).

## In-gel digestion of proteins

In-gel protein digestion was performed as described by Shevchenko et al.[41]. Selected protein bands were excised from the gels, cut into smaller pieces, and destained with 50 mM ammonium bicarbonate in 50% acetonitrile. Proteins were reduced with 10 mM DTT in 100 mM ammonium bicarbonate at 56 °C for 30 min and then alkylated with 50 mM iodoacetamide in 100 mM ammonium bicarbonate in the dark for 30 min. Digestion was performed overnight with 12 ng/µL trypsin (Promega) in 50 mM ammonium bicarbonate at 37 °C. The peptides were extracted with five volumes of acetonitrile; subsequently, the gel pieces were incubated for 5 min in 5% formic acid, and the extraction of peptides was repeated. The eluates were pooled and dried, and the peptides were dissolved in 0.1% trifluoroacetic acid and 2% acetonitrile.

## Liquid chromatography-coupled mass spectrometry

For liquid chromatography-coupled mass spectrometry on a Vanquish Neo system (Thermo Scientific) and Orbitrap Exploris 240 mass spectrometer (Thermo Scientific), peptides were loaded onto a Pep-Map Neo C18 trap column (300 µm × 5 mm, 5 µm particle size, Thermo Scientific) and separated with an EASY-Spray PepMap Neo analytical column with an integrated nanospray emitter (75 µm × 500 mm, 2 µm particle size, Thermo Scientific). Two consecutive linear gradients

were applied at a flow rate of 250 nl/min: 2%–24% solution B for 50 min and 24–40% solution B for 10 min. The two mobile phases used were 0.1% formic acid (v/v) (A) and 80% acetonitrile (v/v) with 0.1% formic acid (B). Eluted peptides were sprayed directly into the mass spectrometer. Precursors were measured in the mass range 350–1,700 m/z with a resolution of 120,000 and selected for fragmentation in a data-dependent mode using the cycle time strategy (2 s) with a dynamic exclusion of 60 s. Higher-energy collisional dissociation (HCD) fragmentation was performed with a normalized collision energy of 30%, and tandem mass spectrometry (MS/MS) scans were conducted with an isolation window of (m/z) 2 and a resolution of 30,000.

## Analysis of MS data
The obtained datasets were processed by MaxQuant[42], versions 2.7.0.0 with the built-in Andromeda search engine with the following parameters: 10 ppm mass tolerance both on precursor and fragment levels, 2 trypsin miscleavages, 1% FDR, carbamidomethylation (C) as a fixed modification, and oxidation (M) as variable modification. The searched database consisted of amino acid sequences of recombinant Wzm and Wzt ΔGH protein sequences from *M. abscessus*, as well as the *E. coli* reference proteome downloaded on 04/07/2024 from UniProt.

## Reporting summary
Further information on research design is available in the Nature Portfolio Reporting Summary linked to this article.

## Data availability
Structural models have been deposited to the Protein Data Bank (PDB) with the accession codes 9QFX (apo Wzm-Wzt*Mabs* in LMNG), 9QGU (ADP-bound Wzm-Wzt*Mabs* in LMNG), 9QH1 (ATP-bound Wzm-Wzt*Mabs* in LMNG), 9QHJ (ATP-bound E178Q Wzm-Wzt*Mabs* in LMNG), 9QHK (apo Wzm-Wzt*Mabs* in nanodiscs), 9QHV (ADP-bound Wzm-Wzt*Mabs* in nanodiscs), 9QHW (ATP-bound Wzm-Wzt*Mabs* in nanodiscs) and 9QHX (ATP-bound E178Q Wzm-Wzt*Mabs* in nanodiscs). The cryo-EM data has been deposited in the Electron Microscopy Data Bank (EMDB) with the accession codes EMD-53123, EMD-53148, EMD-53152, EMD-53171, EMD-53172, EMD-53180, EMD-53181 and EMD-53182 respectively. Plasmids and other data that support the findings of this study are available from the corresponding authors upon request. Source data are provided with this paper.

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

## Acknowledgements

We thank Jana Korduláková and all members of the Seeger lab for stimulating discussions. Work in the laboratory of K.M. was funded by the EU NextGenerationEU through the Recovery and Resilience Plan for Slovakia under the project No. 09I01-03-V03- 00001. Work in the laboratory of M.A.S. was funded by the European Research Council (ERC) (consolidator grant n° 772190), a Sinergia grant by the Swiss National Science Foundation (CRSII--222742) as well as a project grant of the Swiss National Science Foundation (310030_215138). A.A.G. received support from the Forschungskredit of the University of Zurich (FK-21-041). T.L.L. is grateful for the support of GlycoNet (project SD-1) and the Natural Sciences and Engineering Research Council of Canada (project RGPIN-2013-04365). The Center for Microscopy and Image Analysis (ZMB) of the University of Zurich is acknowledged for access to the electron microscope and we thank Dr. Piotr Szwedziak and Dr. Imre Gonda for their technical support. Dr. Peter Baráth and Dr. Maksym Danchenko from the Institute of Chemistry, Slovak Academy of Sciences are acknowledged for performing the mass spectrometry analyses.

## Author contributions

K.M., and M.A.S. conceived the project. K.S. cloned the mycobacterial Wzm-Wzt genes, established the protein purification protocols with the support of A.A.G., K.M., and M.A.S. A.A.G. and V.F. established the nanodisc reconstitution. A.A.G. prepared samples for cryo-EM, collected and analyzed the data, built the Wzm-Wzt$_{Mabs}$ models and refined them. A.A.G., and M.A.S. designed mutation variants with input from K.M. V.F. generated variants and together with K.S., and S.H. performed functional assays in *M. smegmatis* under the supervision of K.M. S.H. performed ddPCR on *M. smegmatis* strains. X.X., and T.L.L. synthesized the substrate analog. A.A.G., V.F., K.S., S.H., K.M., and M.A.S. interpreted the data. A.A.G. prepared the figures, and A.A.G., K.M., and M.A.S. wrote the manuscript with input from all other authors.

## Competing interests

The authors declare no competing interests.
