## [Transparent Peer Review file · Nature Communications]

Structural basis of lipid-linked galactan export by the mycobacterial ABC transporter Wzm-Wzt

Corresponding Author: Professor Markus Seeger

Version 0:

Reviewer comments:

Reviewer #1

(Remarks to the Author)

The manuscript presents a series of eight cryo-EM structures of the ABC-transporter Wzm-Wzt from *Mycobacterium abscessus*, which translocates lipid-linked galactan, a precursor of arabinogalactan, from the cytoplasmic to the periplasmic side of the plasma membrane in mycobacteria. The authors present structures of the Wzm-Wzt complex solubilized with LMNG or embedded in nanodiscs in four different forms (apo; ADP-bound and ATP-bound under turnover conditions; and ATP-trapped using a single-point mutation in the Walker B motif). The authors additionally use three assays to functionally dissect the effect of single-point mutations in the function of the transporter.

The cryo-EM structures show two major arrangements with the nucleotide binding domain being either splayed apart (in the apo and ADP-bound states) or joined together (in the ATP-bound and the ATP-trapped states). The nanodisc structures are very similar in overall fold and arrangement, though fine details are different (NBDs less extended in nanodisc structures, TMD twisted by 10°).

Overall, the study is presented well and the experimental workflows for cryo-EM are robust and analyzed in sufficient detail. Some points for further improvement are noted below:

1) For model validation, only a MolProbity score and Clashscore is provided on the structural statistics table. Adding some model-to-map fit metric should also be included. Possible metrics that should be considered: the Q score that is included in the PDB validation report, the CC score(s) that can be obtained from the cryo-EM validation in Phenix, or an EM ringer score. A description of the score(s) used should be included in the methods section.

2) In all cases where densities for the GH region are shown in the NBD open configuration, two representations of the GH region are shown with one always severely lacking in density to support the overlaid model. Is this presenting the two different protomers of the transporter, and if so, is there a specific reason that one protomer appears to have significantly weaker density in all cases in the NBD open configuration (i.e. Sup fig. 2,4,10).

3) The density for the nucleotide is shown in many different orientations, making comparisons of densities difficult. Also, a figure showing the nucleotide density within the context of the NBD is recommended. In addition, are densities for the nucleotides similar or different on the two protomers?

Reviewer #2

(Remarks to the Author)

Garaeva et al. present a study detailing cryo-EM structures of the lipid-linked galactan transporter Wzm-Wzt from *Mab*. Mycobacterial Wzm-Wzt is an essential ABC transporter that transports an arabinogalactan precursor across the plasma membrane for incorporation into the cell wall. This process is an attractive target for the development of antibiotics. The authors present six structures in different functional states (apo, ADP, and ATP) and in two membrane mimetics (LMNG and nanodisc). A structure of a substrate analogue bound to the transmembrane domain provides the basis for a proposed mechanism of substrate transport. Overall, the paper presents a nice series of structures with supporting mutational data that warrant publication in Nature Communications; however, there are some issues that need addressing prior, most significantly around the accuracy and consistency of the refined models as detailed below.

Major points

1. Being able to inspect the maps/models is appreciated. However, there are some issues that need to be addressed prior to publication: there are examples of incorrectly modeled conformations or inconsistencies between models in regions that are discussed as important to function that, given the resolutions, are not justified. In general, unless the map clearly justifies a difference, then states that appear very similar, eg LMNG wt and EQ, should be modeled consistently.

Examples (not exhaustive):

- In the LMNG wt ATP structure, the ATP-Mg is modeled inconsistently with the other structures here and ABC transporters in general. In the LMNG wt ADP structure, an Mg is included and in a similar position, which is not included in the ADP nanodisc model. Possibly as a result, the adenosine does not fit the density very well and the interaction with the F13 side chain has deviated from the typical parallel stacking interaction.
- In the LMNG wt ATP structure, R61 (side chain), Y62 (side chain), and R63 (backbone) are modeled inconsistently with the EQ LMNG structure. The density better supports the EQ model. This is especially important as this region is proposed to be central to function and the wt model is shown in Fig 1b.
- Given the C2 symmetry, strict NCS should be applied to the final model and there should be no difference in symmetry related chains. This is true for some models/chains but not others.
- For the GH in the closed NBD conformations, in addition to the interaction with the A-loop sheet, G32 of the GH appears to form a direct interaction with ATP that warrants mention. However, there is a difference in how G32 and surrounding residues are modeled in the different structures, notably a difference in backbone orientation of G32 and region around F27. The density for this region (K24-G33) is not well resolved but is the same eg between wt and EQ LMNG so the model should be consistent.
- C-terminal carboxylates should be included in the chains where appropriate. Eg for Wzm Val295, this appears to form an important interaction for folding and possibly function.

2. Authors should update introduction and discussion to include recent publication of *S. aureus* TarGH structures.

3. The use of ATP turnover conditions to study the physiological states is a nice approach. However, it does complicate data processing and interpretation of the regions that are inherently dynamic in both Wzm and Wzt. Did the authors try to obtain structures with eg ADP+substrate, substrate alone, or ATP analogs? This might provide a more convincing substrate bound structure (see below).

4. In line with point 3, is WzmWzt active (ATPase) in LMNG and ND? Given the structures were determined from ATP turn over conditions, then in vitro testing of ATPase activity for the wt and mutants (+/- substrate) would seem straight forward and a valuable addition for wt and mutants. Specifically, is the delta GH mutant active given the apparent role of this region in ATP binding. This mutant in particular seems a very disruptive deletion and I am not sure much can be concluded about the role of the GH from the effect on function.

5. As the authors state, obtaining a substrate bound structure is critical to understanding the export mechanism for this family of transporters and this is currently lacking for related transporters. However, given the resolution, I am not convinced the modeled conformation can be described as unambiguous (line 114). For example, the orientation with hydrophobic prenyl inserted into the cavity is opposite in polarity to both LMNG and the PE (although this is also not completely convincing), which both have their amphiphilic portion inserted and hydrophobic region on the outside. Indeed, in the discussion it is mentioned that "Extensive aromatic interactions with translocating sugar chains are a common theme in polysaccharide transport machineries", which is more like the observed interaction of the LMNG maltose group with the aromatic side chains of F217 and F218. Further, the modeled pyrophosphate binding site, a likely major determinant of specificity in these transporters, also does not seem physiological. Despite the authors assertion (line 297-298), there are positively charged residues lining the cavity that create a significant electropositive surface that would seem more likely to bind phosphate as is observed in the KpsMT structures. The orientation of this substrate, with prenyl inserted first, is the basis for the proposed mechanism and as such should be more confidently supported by the data. Either the conclusions from this structure (and accompanying mechanistic schematic) should be toned down or the data supporting it improved.

6. The differences between the LMNG and nanodisc states are interesting. The authors say the TMD-dimer is more compact in nanodiscs (line 317). Providing the interface surface area and perhaps a comparison of the differences in interactions between the two structures could support this. For example, it appears in nanodiscs that the close interaction between TM1 and TM5 at the cytosolic end is disrupted, specifically the loss of interaction between strictly conserved W71 and D204, this interaction is observed in both WzmWztAa and TarGH and is likely critical to function.

Minor points

1. The data processing figures need more detail in the legends. For example, in Sup Fig 9b and e, and 12c and h, what do the I and II mean? In the processing for the substrate bound structure (Sup Fig 9b; wt ND ADP), in the focussed 3d classification of the open state, did you also observe a class with bound PE like in the apo structure? If not, can you exclude the possibility that the class with something bound is not a mix of substrate and PE?

2. Prime symbol to denote opposite chain should be ' and not an apostrophe (').

Reviewer #3

(Remarks to the Author)

Garaeva, et al report the structural and functional characterization of Wzm-Wzt from *Mycobacterium abscessus*, an ABC transporter responsible for the flipping of lipid-linked galactan (LLG) across the plasma membrane. The authors present a comprehensive series of Wzm-Wzt structures: in nanodiscs vs in detergent; without ATP vs under turnover conditions; and with vs without an LLG analogue. They also carried out structure-function studies using assays based on mycobacterial cell growth and lipid/LLG biochemical analysis.

Type V ABC transporters like Wzm-Wzt are one of the most recent to be structurally characterized, and only a handful of homologous structures have been described to date. Thus, this manuscript helps to expand our knowledge of this relatively

unstudied group of ABC transporters. It also shed some light on some key, unusual features of type V, such as the gate helix (GH), LG-loop, and potential substrate binding residues. Overall, the manuscript is clear and well written, logical, and supported by clear figures. I have a few comments and areas for clarification, which I include below. In particular, I am a little uncertain about the ligand modeling, which may reflect confusion on my part, or could be resolved by softening the conclusions and interpretations somewhat. Overall, I feel this is a very solid manuscript, that just needs some modest revision before acceptance.

MAJOR COMMENTS:

Fig 2j: This panel appears to be a composite of multiple TLC plates (or the same plate with brightness/contrast adjusted differently in different regions?). I have the impression that the F13A plate appears to have more sample loaded, been exposed for longer, or had the brightness/contrast adjusted differently to emphasize weaker bands (e.g., the Glc bands are much darker for F13A compared to other samples). Consequently, it is hard to compare to the WT to see if a faint band for Gal is present there as well. Assuming this was all one plate but that the F13A mutant was just normalized differently to show the weak band better, perhaps it would be appropriate to 1) note this in the figure legend and 2) show in the supplement the entire plate normalized as for the WT, and the entire plate normalized as for F13A, so a reader can easily compare across samples. Perhaps it may also be possible to quantify the signal in each band and present the quantification as well? Fig. 3g has a similar issue.

Ligand modeling: The density for the bound lipids/substrates/detergents in the various structure appears to be significant, though somewhat ambiguous because it does not fully account for the size of the bound LLG-substrate and the "shape complementarity" between the model and the map is modest. I feel that in general, the authors were appropriately cautious in their wording. However, one aspect of the ligand fitting gives me pause, and makes me question whether the maps are insufficient to place these ligands reliably. In short, the overall pose of the LLG-substrate appears different to me than the poses for the LMNG and the PE (I don't have access to the coordinates or maps, this is based only on my interpretation of the figures). It appears to me that for the LLG-substrate, the hydrophobic tail is directed into the inward facing cavity, projecting upwards towards F217/F218/W109, and that the polar head group lies closer to W71 and the NDB. However, for PE, it appears that the polar head group is in the inward facing cavity, and one tail extends downward past W71 and into what appears to be a hydrophilic region near K24 and R61. LMNG also puts a polar group in the cavity (like the PE model), where the LLG-substrate puts a hydrophobic tail. If I am indeed understanding the figures correctly, this seems a bit odd. Why would a tail of PE protrude from the membrane and interact with positively charged residues? Might this interpretation of the map be incorrect? And for the part bound in the inward facing cavity, isn't it odd that it could sometimes bind a hydrophobic farnesyl tail, and sometimes the charged head group of PE? Perhaps there is something that is lost in the figures, by first instinct is that the modeling of one or more of these ligands may not be right.

Spot plate assays: Where spot plates are shown (Fig 2h, Sup Fig 7b), I think all of the mutant strains should ideally be spotted on the same plate along with the empty vector and WT controls. Currently, each strain is shown as a separate sub-panel, and so it is not possible to assess whether these come from the same plate/same experiment, or from multiple experiments. To be properly controlled (e.g., to ensure that the media was correctly prepared in each agar plate), a tested mutant needs to be spotted on the same plate as an empty vector and WT control. While simplest if everything is all on one plate, it would also be acceptable to split mutants between multiple plates, as long as each plate has an empty vector and WT control.

MINOR COMMENTS:

Ln 75: I think there is a typo in this line, perhaps it should read "...Wzm-Wzt and TarGH feature a gate helix..."

Replicates: I appreciate that in some figure legends, the authors state the number of replicates shown or that a panel is representative of X replicates. However, this information is missing for some of the other panels, and should be added. For example, it is missing from Fig 2h, i, j; 3f, g.

Ln 239-242: The authors present SEC data for purified Wzm-Wzt mutants to assess whether the engineered mutations affect complex folding/assembly. Did they by chance also assess whether both subunits (Wzm and Wzt) are present in the peak fraction(s)? Because the complex is being purified via a tag on the transmembrane subunit, it is conceivable that deleting GH might lead to misfiling of the NBD but that the TMDs still dimerize and can be purified. Given the size of the TMDs plus detergent micelle, loss of the NBDs might only lead to a modest shift in retention time. Based on the traces in Sup Fig 6g, the main peak appears to elute slightly before 11 mL, while deltaGH appears to be shifted to ~12 mL. An assay like SDS-PAGE or Western blotting of the peak fractions would provide more clarity on whether both subunit subunits are present and could therefore be much more convincing.

Ln 301-305: See my comment on ligand modeling. I feel like this discussion of broad substrate specificity is a bit speculative given my concerns about the potential unreliability of ligand poses.

Version 1:

Reviewer comments:

Reviewer #1

(Remarks to the Author)

The authors have addressed my previous critiques adequately. I have no further points to make and I believe the manuscript is ready for publication.

Reviewer #2

(Remarks to the Author)

In this revised manuscript, the authors have addressed my concerns mostly around the cryoEM structures and model interpretation. While I believe there is still some ambiguity over the interpretation, this has been acknowledged in the manuscript and alternative explanations proposed. One suggestion prior to publication would be to explain the lack of measured ATPase activity in either LMNG or nanodisc using the malachite green assay, which is at odds with the turn over conditions of some of the structures.

Reviewer #3

(Remarks to the Author)

I appreciate the authors detailed response to the issues I raised in my critique, and most of my concerns have been resolved. However, I feel that two significant issues are not adequately addressed.

1) The spot plate assays: I appreciate that the authors have now shown all of the plates to us reviewers, and everything appears to me to be adequately controlled. But unless I missed it, this data is still not in the manuscript for a reader to assess for themselves. This reduces the transparency and rigor of the final manuscript, as key controls are not being presented and preserved in the scientific record. I think the best solution is to repeat the experiments once more, spotting the strains on a single plate, arranged as the authors want to present them in the final figure. Then there need be no cropping, and there will be no question in a readers mind. Alternatively, and more complicated, would be to include all of the original plates (as provided in the response to reviewers document) in a supplementary figure, so that for each mutant/strain tested, a reader can see the controls from that experiment.

2) Ligand modeling: Reviewer 2 and myself both raised similar concerns about the modeling of the ligands, and I still am concerned that one or more ligands may be modeled incorrectly. Why am I concerned? First, there is the question of the IDENTITY of the ligand(s) responsible for the observed density. This could be any molecule present in the cells the protein was isolated from; any component of the buffers used and anything else knowingly added to the sample; and also anything the sample was unintentionally exposed to (e.g., impurities in chemicals used, chemicals leaching from plastics used, etc). See PMID: 20944227. That is a lot of possibilities to narrow down, even if we just limit ourselves to considering cellular lipids, and anything the protein was exposed to in vitro. The density could result from the superposition of multiple things, not just one. Then there is the binding pose, which is also often tough at these sorts of resolutions. So agreement between the density and the ligand, and the chemical plausibility of the interaction are key factors. In this case, I feel both are in question. The discrepancy between the binding poses of the 3 different proposed ligands and the interactions between the ligands and the protein raise questions about chemical plausibility. The fact that the entire molecules don't fit into the observed density, and the modest complementarity between density and modeled ligands, raises questions about the whether the assignment of their identity is correct. Numerous molecules could be docked into this density with fits that are similar quality to the modeled LLG. That says that the map is not providing much information about the identity or pose of the the bound molecule(s). The authors may be right, perhaps part of LLG is disordered, and that is why the map can only explain a part of it; however, it seems very plausible to me that it simply is not LLG, or the density is an average with contributions from LLG along with other lipid and detergent. I do not feel that the evidence is strong enough to assign the identity of these ligands, when they are a major crux of the manuscript. Perhaps these ligands should be deposited using the "unknown ligand" code UNL? Consequently, I still believe that speculating on the promiscuity of the ligand binding site lacks rigor, when the identity of the bounds ligands is in question--one can't begin to discuss how such diverse ligands are bound, if the evidence supporting their binding to a particular site with a particular pose is weak.

Version 2:

Reviewer comments:

Reviewer #2

(Remarks to the Author)

The acknowledgement of the ambiguous quality of the ligand density and marking of the PDB as such (UNL) is appreciated and important for future users of the coordinates. Not clear that the timeframe of the cryoEM incubation prior to vitrification and the assay conditions for the malachite green assay are so different that one would show turnover and the other not. However I am willing to accept the paper with the current revisions.

Reviewer #3

(Remarks to the Author)

I appreciate the authors responsiveness to my concerns. They are now resolved and I strongly support acceptance. Nice manuscript!

We thank the reviewers for their careful, comprehensive and constructive assessment of our work.

The most important changes of the manuscript are:

1. Using our functional assays, we analyzed two additional mutations of Wzm-Wzt_{Mabs}, namely R61A^{Wzm} and K24E^{Wzt}, which support substrate analog position in the binding site with the farnesyl group inserted first and provide further importance to the GH element in NBD in LLG translocation. This data is shown in Fig. 3 and Supplementary Figs. 6-8.
2. We extended our analysis of differences in TMD orientation and interaction between ATP-bound Wzm-Wzt_{Mabs} structures in LMNG and nanodiscs. The results are summarised in updated Supplementary Fig. 17 and in the main text.
3. We reanalyzed ATP, ADP and magnesium binding in our structures and improved Supplementary Figs. 2, 4, 5, 11 and 13.
4. We revisited ATP-turnover dataset in nanodiscs and after careful examination of each class resulting from focused 3D classification, we observed a PE-bound class present in this dataset, which separated from the substrate analog-bound class. We modified Supplementary Fig. 10b and now mention this finding in the main text.
5. We addressed the raised issues concerning the proper controls in the spot assays (Fig. 2h, Supplementary Fig. 7b); number of replicates (Figs. 2h-j, 3f-g, Supplementary Fig. 8a-c) and presentation of TLC analyses of the sugar composition (Figs. 2j, 3g, Supplementary Fig. 8b) in the E-soak extracts in the methods section, and/or in the figure legends, respectively. All raw data, including the uncropped TLC plates and complete agar plates are presented in the Source Data xls file.
6. We added two new supplementary figures: Supplementary Fig. 9 shows mass spectrometry data of the purified Wzm-Wzt_{Mabs} Δ GH mutant. Supplementary Fig. 12 features a detailed analysis of two possible poses of LLG-substrate analog. Due to these addition numbering of the original supplementary figures was changed.

We have carefully edited the manuscript to address all reviewers' concerns and to improve clarity of text and figures.

Reviewer #1 (Remarks to the Author):

The manuscript presents a series of eight cryo-EM structures of the ABC-transporter Wzm-Wzt from *Mycobacterium abscessus*, which translocates lipid-linked galactan, a precursor of arabinogalactan, from the cytoplasmic to the periplasmic side of the plasma membrane in mycobacteria. The authors present structures of the Wzm-Wzt complex solubilized with LMNG or embedded in nanodiscs in four different forms (apo; ADP-bound and ATP-bound under turnover conditions; and ATP-trapped using a single-point mutation in the Walker B motif). The authors additionally use three assays to functionally dissect the effect of single-point mutations in the function of the transporter. The cryo-EM structures show two major arrangements with the nucleotide binding domain being either

splayed apart (in the apo and ADP-bound states) or joined together (in the ATP-bound and the ATP-trapped states). The nanodisc structures are very similar in overall fold and arrangement, though fine details are different (NBDs less extended in nanodisc structures, TMD twisted by 10°). Overall, the study is presented well and the experimental workflows for cryo-EM are robust and analyzed in sufficient detail.

We thank reviewer #1 for this positive assessment of our work.

Some points for further improvement are noted below:

1) For model validation, only a MolProbity score and Clashscore is provided on the structural statistics table. Adding some model-to-map fit metric should also be included. Possible metrics that should be considered: the Q score that is included in the PDB validation report, the CC score(s) that can be obtained from the cryo-EM validation in Phenix, or an EM ringer score. A description of the score(s) used should be included in the methods section.

We added Q score and CC score to the Supplementary Table 1 with validation statistics. The description of the scores and how they were calculated is included in the methods section.

2) In all cases where densities for the GH region are shown in the NBD open configuration, two representations of the GH region are shown with one always severely lacking in density to support the overlaid model. Is this presenting the two different protomers of the transporter, and if so, is there a specific reason that one protomer appears to have significantly weaker density in all cases in the NBD open configuration (i.e. Sup fig. 2,4,10).

We apologise for not being clear with these figures. All cryo-EM maps and corresponding models reported in this study have C2 symmetry, therefore densities for both protomers are identical. The original figures showed GH from the same protomer at two different counter levels. To avoid confusions, we modified these figures (Supplementary Figs. 2, 4, 5, 11 and 13) and kept only one GH representation with a counter level allowing to see full density of the GH.

3) The density for the nucleotide is shown in many different orientations, making comparisons of densities difficult. Also, a figure showing the nucleotide density within the context of the NBD is recommended. In addition, are densities for the nucleotides similar or different on the two protomers?

The densities for nucleotides are identical as the maps were processed with C2 symmetry. We corrected the respective figures (Supplementary Figs. 4, 5, 11 and 13) now showing nucleotides in a similar orientation and included panels showing the nucleotide density within the context of the NBD.

Reviewer #2 (Remarks to the Author):

Garaeva et al. present a study detailing cryo-EM structures of the lipid-linked galactan transporter Wzm-Wzt from *Mab*. Mycobacterial Wzm-Wzt is an essential ABC transporter that transports an arabinogalactan precursor across the plasma membrane for incorporation into the cell wall. This process is an attractive target for the development of antibiotics. The authors present six structures in different functional states (apo, ADP, and ATP) and in two membrane mimetics (LMNG and nanodisc). A structure of a substrate analogue bound to the transmembrane domain provides the basis for a proposed mechanism of substrate transport. Overall, the paper presents a nice series of structures with supporting mutational data that warrant publication in *Nature Communications*; however, there are some issues that need addressing prior, most significantly around the accuracy and consistency of the refined models as detailed below.

We thank reviewer #2 for this positive assessment of our work and for the careful inspection of the models and the maps.

Major points

1. Being able to inspect the maps/models is appreciated. However, there are some issues that need to be addressed prior to publication: there are examples of incorrectly modeled conformations or inconsistencies between models in regions that are discussed as important to function that, given the resolutions, are not justified. In general, unless the map clearly justifies a difference, then states that appear very similar, eg LMNG wt and EQ, should be modeled consistently.

We thank the reviewer for spotting inconsistencies and mistakes in our models. In the course of the revision we have addressed all concerns and the models have been corrected.

Examples (not exhaustive):

- In the LMNG wt ATP structure, the ATP-Mg is modeled inconsistently with the other structures here and ABC transporters in general. In the LMNG wt ADP structure, an Mg is included and in a similar position, which is not included in the ADP nanodisc model. Possibly as a result, the adenosine does not fit the density very well and the interaction with the F13 side chain has deviated from the typical parallel stacking interaction.

We thank reviewer#2 for this comment. The Mg-ion is now placed correctly in LMNG wt ATP structure, also in agreement to other ABC transporter structures.

After inspecting our ADP-bound structures and comparing our data to high-resolution crystal structures of ADP-bound ABC transporters, we concluded that the density does not support the modelling of Mg into structures. The position of F13 parallel to adenosine has been checked and adjusted.

- In the LMNG wt ATP structure, R61 (side chain), Y62 (side chain), and R63 (backbone) are modeled inconsistently with the EQ LMNG structure. The density better supports the EQ model. This is especially important as this region is proposed to be central to function and the wt model is shown in Fig 1b.

We thank reviewer#2 for spotting this error. We corrected the models such that they are now consistent.

- Given the C2 symmetry, strict NCS should be applied to the final model and there should be no difference in symmetry related chains. This is true for some models/chains but not others.

Thank you for spotting this mistake. We corrected all models reported in this study accordingly.

- For the GH in the closed NBD conformations, in addition to the interaction with the A-loop sheet, G32 of the GH appears to form a direct interaction with ATP that warrants mention. However, there is a difference in how G32 and surrounding residues are modeled in the different structures, notably a difference in backbone orientation of G32 and region around F27. The density for this region (K24-G33) is not well resolved but is the same eg between wt and EQ LMNG so the model should be consistent.

We carefully inspected both models in the region around G32 and rebuilt them such that the ATP-bound wild-type and EQ structures are consistent. Interaction of G32 with ATP is now mentioned in the text in line 187: As a consequence, the backbone carbonyl of the GH-residue G32^{Wzt} appears to establish a direct interaction with the ribose hydroxyl group of ATP (Fig. 2f).

- C-terminal carboxylates should be included in the chains where appropriate. Eg for Wzm Val295, this appears to form an important interaction for folding and possibly function.

We thank the reviewer for spotting this important detail, which we have corrected.

2. Authors should update introduction and discussion to include recent publication of *S. aureus* TarGH structures.

The respective paper is cited in the revised version of the manuscript.

3. The use of ATP turnover conditions to study the physiological states is a nice approach. However, it does complicate data processing and interpretation of the regions that are inherently dynamic in

both Wzm and Wzt. Did the authors try to obtain structures with eg ADP+substrate, substrate alone, or ATP analogs? This might provide a more convincing substrate bound structure (see below).

We agree with the reviewer that ATP turnover conditions can complicate data processing due to conformational heterogeneity. However, this approach is now widely used to capture the physiological range of conformations in ABC transporters within a single sample. In our hands, ATP hydrolysis appeared to proceed slowly enough to allow resolution of both ATP-bound and ADP-bound states in the same dataset.

The reduced resolution in the ADP-bound states primarily reflects the increased flexibility and separation of the NBDs, which is expected for this conformation. While we did not pursue conditions with ADP+substrate or substrate alone in this study, we believe that the 3.7–4.0 Å resolution achieved in the ADP-bound structures is sufficient to support our key conclusions about substrate positioning and conformational changes.

Regarding ATP analogs, we have found them to be relatively ineffective at promoting full NBD closure in this system. For that reason, we opted to use the catalytically impaired E178Q mutant to obtain a more stable, ATP-trapped state.

4. In line with point 3, is WzmWzt active (ATPase) in LMNG and ND? Given the structures were determined from ATP turn over conditions, then in vitro testing of ATPase activity for the wt and mutants (+/- substrate) would seem straight forward and a valuable addition for wt and mutants. Specifically, is the delta GH mutant active given the apparent role of this region in ATP binding. This mutant in particular seems a very disruptive deletion and I am not sure much can be concluded about the role of the GH from the effect on function.

We attempted to measure the activity of wild-type Wzm-Wzt versus the E178Q mutant in both LMNG and nanodiscs using a Malachite Green assay. However, we did not observe any significant difference between the wild-type and catalytically inactive mutant, likely due to the very slow rate of ATP hydrolysis in vitro, which makes detection with this assay (and any other ATPase activity assay) challenging. We also tried to see whether ATPase activity can be stimulated with our LLG-analogue, which was unfortunately not the case in the concentration range we were able to test (we had only a very small amount of LLG-analogue available). But indeed, as the reviewer points out, the fact that we see both the ATP- and ADP-bound conformation under ATP-turnover conditions strongly suggests that purified Wzm-Wzt exhibits (slow) ATPase activity. From a physiological point of view, it is plausible to assume that ATPase activity of Wzm-Wzt is tightly coupled in vivo, and ATP is only hydrolyzed at a rapid rate once the natural substrate has entered the channel.

Regarding the Δ GH mutant, we concur with the reviewer that this is a rather drastic alteration, and indeed, it is possible that the observed loss-of-function phenotype could be due to impaired ATP hydrolysis (which we unfortunately cannot experimentally address owing to the overall slow

hydrolysis rate of purified Wzm-Wzt). To further investigate the role of the GH region, we introduced a more subtle point mutation, K24E^{Wzt}, which alters a residue located near the substrate-analog binding site in our ADP-bound nanodisc structure. This mutation likely affects interactions with the negatively charged diphosphate moiety of LLG. In our functional assays the K24E variant exhibits a growth defect in liquid media, increased production of TMM and TDM, as well as radioactive Gal accumulation supporting the notion that the GH region is critical for the substrate loading or translocation. These results have been added to the revised manuscript (line 325 ff, and Figs. 3 and Supplementary Figs. 6-8).

5. As the authors state, obtaining a substrate bound structure is critical to understanding the export mechanism for this family of transporters and this is currently lacking for related transporters. However, given the resolution, I am not convinced the modeled conformation can be described as unambiguous (line 114). For example, the orientation with hydrophobic prenyl inserted into the cavity is opposite in polarity to both LMNG and the PE (although this is also not completely convincing), which both have their amphiphilic portion inserted and hydrophobic region on the outside. Indeed, in the discussion it is mentioned that “Extensive aromatic interactions with translocating sugar chains are a common theme in polysaccharide transport machineries”, which is more like the observed interaction of the LMNG maltose group with the aromatic side chains of F217 and F218. Further, the modeled pyrophosphate binding site, a likely major determinant of specificity in these transporters, also does not seem physiological. Despite the authors assertion (line 297-298), there are positively charged residues lining the cavity that create a significant electropositive surface that would seem more likely to bind phosphate as is observed in the KpsMT structures. The orientation of this substrate, with prenyl inserted first, is the basis for the proposed mechanism and as such should be more confidently supported by the data. Either the conclusions from this structure (and accompanying mechanistic schematic) should be toned down or the data supporting it improved.

We wish to thank the reviewer for pointing out these important points.

First, the density we assign to the LLG analogue is unique for this dataset and does not appear in the control density obtained without adding the LLG analogue. In fact, this is what we meant with unambiguous density in line 114 in the initial version of the manuscript. We agree that this wording might be misunderstood, hence we reformulated the sentence in the following manner:

Line 114: “..., but we only observed density attributable to this substrate when the transporter was prepared in nanodiscs (see below).”

In response to this comment, we performed a more thorough interpretation of the LLG analogue density. We agree with reviewer#2 (and also with reviewer#3) that the quality of the density leaves room for alternative interpretations.

We prepared a new Supplementary Fig. 12 containing an analysis of two possible orientations of the LLG analogue in the substrate binding site: the “lipid first” and “sugars first” poses. Based on visual inspection of the cryo-EM density, interaction analysis and the analysis of additional mutations, we favor the “lipid first” orientation of the LLG analogue. Here are the reasons for this interpretation:

1. Both poses of the LLG-analogue (“lipid first” and “sugars first”) fit into the cryo-EM density reasonably well (see Supplementary Figure 12 a-b). However, the density inside of the TMD is relatively narrow – supporting the fit of the prenyl moiety (in the “lipid first” pose) better than that of two galactofuranoses (in the “sugars first” pose). Given the size and flexibility of sugars, we would expect a broader density more similar to what we observe for LMNG. A side-by-side comparison of the LMNG density (apo_LMNG map) with both LLG poses is shown for reference:

2. The phosphate group typically produces strong, well-resolved density in cryo-EM due to its high electron-scattering properties. In the “lipid first” orientation, the diphosphate group aligns with strong density, consistent with this expectation. In contrast, the diphosphate appears poorly resolved in the “sugars first” pose—likely reflecting flexibility, but still supporting “lipid first” as the more plausible fit.
3. In the “sugars first” orientation, R61^{Wzm} in the LG-loop would establish a direct interaction with the diphosphate (see 2D representations of LLD binding in Supplementary Figure 12 f). This arginine, present in both KpsMT (R35) and in Wzm-Wzt_{A0} (R17), was shown to be critical for KpsMT, because its mutation to alanine disrupted capsular polysaccharide export (Kuklewicz, J., Zimmer, J. *Nature* 2024). We mutated R61^{Wzm} in Wzm-Wzt_{Mabs} to alanine, but did not observe any effect on LLG transport in our functional assays (Fig. 3f-g; Supplementary Figs. 7b and 8c).
4. In the “lipid first” pose, K24^{Wzt} in the GH of the NBD forms a hydrogen bond with GlcNAc (see 2D representations of LLD binding in Supplementary Figure 12 e) and is well-positioned to interact with the negatively charged diphosphate moiety of LLG. Mutation of K24^{Wzt} to

glutamate showed a clear growth defect in liquid culture, accumulation of TDM, TMM and of radioactively labelled galactose (Fig. 3f-g; Supplementary Figs. 7c, 8a, c), which are hallmarks of a non-functional Wzm-Wzt_{Mabs} transporter. The transport defect caused by the K24E^{Wzt} mutation is in favour of the lipid-first binding pose.

5. Transport of LLG by Wzm-Wzt_{Mabs} is known to be coupled with polymerization of the galactan chain by the GlfT2 enzyme (Karin Savková, Stanislav Huszár *et al*, *PNAS* 2021), similar to the production and export of an O2a antigen in *Klebsiella pneumoniae* (Kos, V., L. Cuthbertson and C. Whitfield (2009). "The *Klebsiella pneumoniae* O2a antigen defines a second mechanism for O antigen ATP-binding cassette transporters." *J Biol Chem* 284(5): 2947-2956. DOI: 10.1074/jbc.M807213200). This suggests that the sugar moiety cannot enter the transporter first, further supporting the "lipid first" model.
6. In the KpsMT system, the phospholipid enters the transporter first, with the polysaccharide being threaded through last (Kuklewicz, J., Zimmer, J. *Nature* 2024). KpsMT's proposed transport mechanism is consistent with the "lipid-first" orientation observed in our structure.

Since substrate analog-bound structure of Wzm-Wzt is an important part of this manuscript, we added the following section to the main text (line 264 ff): The cryo-EM density can also be interpreted in an alternative way wherein the Galf-Galf-Rhap-GlcpNAc-P-P-moiety of the substrate analog enters first ("sugar-first" binding pose) (Supplementary Fig. 12). However, we consider the "lipid first" binding pose of the substrate analog to be more likely for the following reasons: (i) the prenyl moiety fits better into the narrow density extending into the TMD than the sugar-chain; (ii) there is stronger density for the diphosphate moiety in the "lipid-first" binding pose versus the "sugar-first" binding pose; (iii) the prenyl chain is accommodated well in the apolar cavity by interacting with W109^{Wzm}, F217^{Wzm} and F218^{Wzm}. In further support of the "lipid-first" pose, it has been shown that the extension of the arabinogalactan chain mediated by the GlfT2 polymerase and substrate translocation are very likely coupled processes⁸, which precludes the possibility that the sugar chain enters first. Finally, an analogous "lipid-first" translocation mechanism has been described for KpsMT²⁴. Hence, from a physiological point of view, it is reasonable to propose that the prenyl-moiety enters first. Nevertheless, we cannot exclude the possibility that the substrate analog entered in the alternative, "sugars first" orientation (Supplementary Fig. 12), which would then represent a non-physiological binding mode.

6. The differences between the LMNG and nanodisc states are interesting. The authors say the TMD-dimer is more compact in nanodiscs (line 317). Providing the interface surface area and perhaps a comparison of the differences in interactions between the two structures could support this. For example, it appears in nanodiscs that the close interaction between TM1 and TM5 at the cytosolic

end is disrupted, specifically the loss of interaction between strictly conserved W71 and D204, this interaction is observed in both WzmWztAa and TarGH and is likely critical to function.

We thank reviewer#2 for this constructive suggestion. We performed an analysis of the interface area and the interactions between TM1 and TM5 and show the results in Supplementary Fig. 17. We also added an additional section to the main text (line 347 ff): The interface surface area between the two TMD subunits, as calculated by PDBePISA, increases from 1361 Å² in LMNG to 1688 Å² in nanodiscs (Supplementary Fig. 17e). Intriguingly, six hydrogen bonds and a salt bridge stabilize the TMD dimer in the structure determined in LMNG (Supplementary Fig. 17f). In contrast, these interactions are disrupted in the nanodisc environment, thereby weakening the contact between TM1 and TM5' of the opposite subunit at the cytosolic end (Supplementary Fig. 17f). Due to these conformational rearrangements, the substrate-binding cavity between TM1 and TM5' changes its shape from a relatively flat surface in LMNG to a convex shape in nanodiscs (Supplementary Fig. 17c-d).

Minor points

1. The data processing figures need more detail in the legends. For example, in Sup Fig 9b and e, and 12c and h, what do the I and II mean? In the processing for the substrate bound structure (Sup Fig 9b; wt ND ADP), in the focussed 3d classification of the open state, did you also observe a class with bound PE like in the apo structure? If not, can you exclude the possibility that the class with something bound is not a mix of substrate and PE?

In the revised manuscript, Supplementary Fig. 9 corresponds to Supplementary Fig. 10 and Supplementary Fig. 12 corresponds to Supplementary Fig. 14.

In Supplementary Fig. 10b and e, and Supplementary Fig. 14c and h, the I and II indicate steps at which refinement with C1 symmetry has been performed to check for potential C2-symmetry artefacts. We modified these figures and figure legends for better clarity.

Regarding the processing of the substrate-bound dataset, the reviewer raises an important point. We revisited this dataset and carefully examined each class resulting from focused 3D classification. One class yielded the final map of the substrate- and ADP-bound Wzm-Wzt complex (blue map in Supplementary Fig. 10b). We also closely examined the second class from this classification (highlighted by a red rectangle – see figure below). This class contained 19,405 particles and, after non-uniform refinement, resulted in a 4.3 Å resolution map. Although the density in the binding site is not well-defined, it appears more consistent with a PE lipid than with the substrate analog (see figures below). This suggests that the PE-bound and substrate-bound particles were likely separated during data processing.

Part of the Supplementary Fig. 10b. The map on the right, highlighted by the red rectangle, might represent the PE-bound state.

Density of the TMD binding site of the "PE-bound?" class (see red frame above) after refinement. For comparison, the structures of the apo-ND with a bound PE lipid (yellow) and the ADP-ND with bound substrate analog (green) were placed into this map. Clearly, a PE lipid fits better into the observed density than the LLG analog. Please note that the overall map quality is rather poor and required contouring at low sigma levels.

This is how the density for the substrate analog looks like in ADP-ND map for the reference.

Same analysis as above, but this time showing the entire density map with purple highlighting non-proteinaceous density. While the density is poorly resolved, we believe it more closely resembles a PE lipid than the substrate analog.

These results are now described in Methods section and depicted in Supplementary Figure 10.

2. Prime symbol to denote opposite chain should be ' and not an apostrophe (').

Corrected.

Reviewer #3 (Remarks to the Author):

Garaeva, et al report the structural and functional characterization of Wzm-Wzt from *Mycobacterium abscessus*, an ABC transporter responsible for the flipping of lipid-linked galactan (LLG) across the plasma membrane. The authors present a comprehensive series of Wzm-Wzt structures: in nanodiscs vs in detergent; without ATP vs under turnover conditions; and with vs without an LLG analogue. They also carried out structure-function studies using assays based on mycobacterial cell growth and lipid/LLG biochemical analysis.

Type V ABC transporters like Wzm-Wzt are one of the most recent to be structurally characterized, and only a handful of homologous structures have been described to date. Thus, this manuscript helps to expand our knowledge of this relatively unstudied group of ABC transporters. It also shed

some light on some key, unusual features of type V, such as the gate helix (GH), LG-loop, and potential substrate binding residues. Overall, the manuscript is clear and well written, logical, and supported by clear figures. I have a few comments and areas for clarification, which I include below. In particular, I am a little uncertain about the ligand modeling, which may reflect confusion on my part, or could be resolved by softening the conclusions and interpretations somewhat. Overall, I feel this is a very solid manuscript, that just needs some modest revision before acceptance.

We thank reviewer#3 for this positive assessment of our work.

MAJOR COMMENTS:

Fig 2j: This panel appears to be a composite of multiple TLC plates (or the same plate with brightness/contrast adjusted differently in different regions?). I have the impression that the F13A plate appears to have more sample loaded, been exposed for longer, or had the brightness/contrast adjusted differently to emphasize weaker bands (e.g., the Glc bands are much darker for F13A compared to other samples). Consequently, it is hard to compare to the WT to see if a faint band for Gal is present there as well. Assuming this was all one plate but that the F13A mutant was just normalized differently to show the weak band better, perhaps it would appropriate to 1) note this in the figure legend and 2) show in the supplement the entire plate normalized as for the WT, and the entire plate normalized as for F13A, so a reader can easily compare across samples. Perhaps it may also be possible to quantify the signal in each band and present the quantification as well? Fig. 3g has a similar issue.

We agree with the reviewer that these issues should be clarified. Both Fig. 2j and Fig. 3g are, indeed, composites of two TLC plates – the samples from mutants with intermediate phenotypes (F13A^{Wzt} in Fig. 2j; and W109A^{Wzm}, F217A^{Wzm} and F218A^{Wzm} in Fig. 3g) were run separately, the rest of the lanes were on one plate, as shown below.

The reason for this approach was to decrease the effects of the high intensity of the bands accumulated due to the lethal mutations, so that the subtle changes caused by mutations with the intermediate phenotypes could be evaluated. For TLCs with the intermediate phenotype mutations, we included as a “standard” just 10 % of the material from the ATc-treated empty plasmid control and these TLC plates were exposed for longer time compared to the plates with lethal mutants to increase the signal-to-noise ratio. The entire plates were adjusted for contrast and the corresponding lanes were extracted afterwards.

Since the signals for Gal in mutants with intermediate phenotypes are rather weak, in the Supplemental Material we included the results for all three biological replicates that were carried out (Supplementary Fig. 8b, right panel), one of which is presented also in the main Figs. 2j and 3g. Each of these mutants show the difference in signals for Gal in +/- ATc pairs. This was not observed for the complementation with the WT version of the transporter at any of the experiments. To emphasize the lack of the signal for Gal in the WT complementants, we included a TLC normalized for WT (+ATc) lane by ImageJ into the modified Supplementary Fig. 8d.

The same comparison for the plates presented in the main Fig. 2j is provided below.

We modified the figure legends to address all issues raised by the reviewer in this comment, as well as in the comment below concerning the number of replicates.

The modified figure legend for Fig. 2i-j.

“i, TLC analysis of lipids extracted from cells grown in the absence or presence of ATc. The lipids were separated on silica gel plates in CHCl₃/CH₃OH/H₂O (20:4:0.5, vol/vol), and visualized with cupric sulfate. A representative image of three biological replicates is shown. TDM—trehalose dimycolates, TMM—trehalose monomycolates, PE—phosphatidyl ethanolamine, and CL—cardiolipin. j, Monosaccharide composition of E-soak extracts of cells grown in radiolabeled medium in the absence or presence of ATc. TLC plates were read out by autoradiography. The F13A^{Wzt} sample was separated on a different TLC plate and exposed for longer to increase the signal-to-noise ratio. Representative images of three biological replicates are shown. Ara—arabinose, Man—mannose, Glc—glucose, Gal—galactose.”

The modified figure Fig. 3 f-g also includes the two added mutations, namely R61A in Wzm and K24E in Wzt. The modified legend for this figure is as follows:

“f, TLC analysis of lipids extracted from *Msmeg CRISPRi-wzt* strains complemented with wild-type or Wzm-Wzt_{Mabs} variants (W109A^{Wzm}-F217A^{Wzm}-F218A^{Wzm}, W71A^{Wzm}, L67R^{Wzm}, R61A^{Wzm}, K24E^{Wzt}) grown in the absence or presence of ATc. TDM—trehalose dimycolates, TMM—trehalose monomycolates, PE—phosphatidyl ethanolamine, and CL – cardiolipin. A representative image of at least two biological replicates is shown. g, Monosaccharide composition of E-soak extracts of cells grown in radiolabeled medium in the absence or presence of ATc. TLC plates were read out by autoradiography. The W109A^{Wzm}-F217A^{Wzm}-F218A^{Wzm} variant, as well as R61A^{Wzm} and K24E^{Wzt} variants were separated on different TLC plates and exposed for longer to increase the signal-to-noise ratio. Representative images of at least two biological replicates are shown. Ara – arabinose, Man – mannose, Glc – glucose, Gal – galactose.”

The expression “at least two” in the legends for Fig. 3f-g refers to the fact, that contrary to the original mutations which were all examined in three biological replicates, the analyses of the two new mutations R61A^{Wzm} and K24E^{Wzt} were performed in two biological replicates. Both are shown in modified Supplementary Fig. 8c.

Ligand modeling: The density for the bound lipids/substrates/detergents in the various structure appears to be significant, though somewhat ambiguous because it does not fully account for the size of the bound LLG-substrate and the "shape complementarity" between the model and the map is modest. I feel that in general, the authors were appropriately cautious in their wording. However, one aspect of the ligand fitting gives me pause, and makes me question whether the maps are insufficient to place these ligands reliably. In short, the overall pose of the LLG-substrate appears different to me than the poses for the LMNG and the PE (I don't have access to the coordinates or maps, this is based only on my interpretation of the figures). It appears to me that for the LLG-substrate, the hydrophobic tail is directed into the inward facing cavity, projecting upwards towards F217/F218/W109, and that the polar head group lies closer to W71 and the NDB. However, for PE, it appears that the polar head group is in the inward facing cavity, and one tail extends downward past W71 and into what appears to be a hydrophilic region near K24 and R61. LMNG also puts a polar group in the cavity (like the PE model), where the LLG-substrate puts a hydrophobic tail. If I am indeed understanding the figures correctly, this seems a bit odd. Why would a tail of PE protrude from the membrane and interact with positively charged residues? Might this interpretation of the map be incorrect? And for the part bound in the inward facing cavity, isn't it odd that it could sometime bind a hydrophobic farnesyl tail, and sometimes the charged head group of PE? Perhaps there is something that is lost in the figures, by first instinct is that the modeling of one or more of these ligands may not be right.

The reviewer is correct in his interpretation of the orientation of LMNG, PE and LLG-analogue in our structures (see also the overview provided in Supplementary Fig. 15). The densities for LMNG and PE permit for the unambiguous modelling of these molecules (see also Fig. 1c for LMNG and Fig. 4a for PE). In contrast, the cryo-EM map of the LLG substrate analog-bound Wzm-Wzt is of lower resolution and as an additional challenge, the observed density does not represent the full length LLG. As part of the revision, we built the LLG-analogue also in its alternative “sugar-first” into the density (Supplementary Fig. 12), side-by-side with the original “lipid-first” pose. Neither the cryo-EM density nor the interaction of the LLG-analogue with side chains can unambiguously rule out the alternative “sugar-first” pose. Nevertheless, there are several arguments speaking for the “lipid-first” pose, as we also outline in our answer to reviewer#2. In addition, we openly discuss the possibility of the alternative binding pose of the LLG analogue (line 264 ff): The cryo-EM density can also be interpreted in an alternative way wherein the Galf-Galf-Rhap-GlcpNAc-P-P-moiety of the substrate analog enters first (“sugar-first” binding pose) (Supplementary Fig. 12). However, we consider the “lipid first” binding pose of the substrate analog to be more likely for the following reasons: (i) the prenyl moiety fits better into the narrow density extending into the TMD than the sugar-chain; (ii) there is stronger density for the diphosphate moiety in the “lipid-first” binding pose versus the “sugar-first” binding pose; (iii) the prenyl chain is accommodated well in the apolar cavity by interacting with W109^{Wzm}, F217^{Wzm} and F218^{Wzm}. In further support of the “lipid-first” pose, it has been shown that the extension of the arabinogalactan chain mediated by the GlfT2 polymerase and substrate translocation are very likely coupled processes⁸, which precludes the possibility that the sugar chain enters first. Finally, an analogous “lipid-first” translocation mechanism has been described for KpsMT²⁴. Hence, from a physiological point of view, it is reasonable to propose that the prenyl-moiety enters first. Nevertheless, we cannot exclude the possibility that the substrate analog entered in the alternative, “sugars first” orientation (Supplementary Fig. 12), which would then represent a non-physiological binding mode.

Spot plate assays: Where spot plates are shown (Fig 2h, Sup Fig 7b), I think all of the mutant strains should ideally be spotted on the same plate along with the empty vector and WT controls. Currently, each strain is shown as a separate sub-panel, and so it is not possible to assess whether these come from the same plate/same experiment, or from multiple experiments. To be properly controlled (e.g., to ensure that the media was correctly prepared in each agar plate), a tested mutant needs to be spotted on the same plate as an empty vector and WT control. While simplest if everything is all on one plate, it would also be acceptable to split mutants between multiple plates, as long as each plate has an empty vector and WT control.

We thank the reviewer for this comment and agree that this needs clarification. The images are from multiple plates. For each strain complemented with the mutant versions of the transporter two

clones were tested along with the appropriate controls; one of the mutant strains was chosen for further experiments. These clones are also shown in the Fig. 2h and Sup. Fig. 7b.

Complete plates for Fig. 2h:

In this figure one clone for each strain was spotted in duplicates. The images used for the figure are highlighted (the figures below show raw images not adjusted for contrast or brightness).

Complete plates for Supplementary Fig. 7b:

In this figure two clones for each strain complemented with the mutant versions of the transporter are shown, along with the appropriate controls included on each plate. Two additional mutants, R61A^{Wzm} and K24E^{Wzt}, were added to Supplementary Fig. 7b – these are labelled with a rectangle. The clones/images used for the figure are highlighted (the figures below show raw images not adjusted for contrast or brightness).

To clarify the results shown in Fig. 2h and Supplementary Fig. 7b we modified the **METHODS** section, chapter **Preparation of the complemented Msmeg CRISPRi-wzt strains and their phenotypic characterization** as follows:

“For an initial evaluation of the growth, **two clones** for the control and complemented *Msmeg* CRISPRi-wzt strains were plated in serial dilutions on solid 7H11 medium supplemented with 10 % (v/v) OADC, kanamycin (20 µg/ml), hygromycin (50 µg/ml) in the absence or presence of ATc (100 ng/ml). **Each plate with mutated pFLAG_OriM-wzt-glfT1-wzm_{Mabs} contained the control strain Msmeg CRISPRi-wzt transformed with the empty pFLAG_OriM and the Msmeg CRISPRi-wzt strain transformed with the wild type version of pFLAG_OriM-wzt-glfT1-wzm_{Mabs}.** Plates were cultivated at 37 °C for 2 – 4 days until visible colonies formed. **The growth of the two clones was comparable in each tested strain and one of them was selected for further experiments.** The growth of these strains in liquid medium was analyzed in glycerol-alanine-salts (GAS) medium supplemented with tyloxapol (0.025 %), kanamycin (20 µg/ml), hygromycin (50 µg/ml) in the absence or presence of ATc (100 ng/ml) at 37 °C and 120 rpm. The media were inoculated with fresh early-log pre-cultures to the initial OD₆₀₀ = 0.01.”

MINOR COMMENTS:

Ln 75: I think there is a typo in this line, perhaps it should read "...Wzm-Wzt and TarGH feature a gate helix..."

Corrected.

Replicates: I appreciate that in some figure legends, the authors state the number of replicates shown or that a panel is representative of X replicates. However, this information is missing for some of the other panels, and should be added. For example, it is missing from Fig 2h, i, j; 3f, g.

We added the information for Fig. 2h in the figure legend and in the **METHODS** section, chapter **Preparation of the complemented Msmeg CRISPRi-wzt strains and their phenotypic characterization**; and information for Fig. 2i,j and 3f,g to the figure legends, as stated above.

The modified figure legend for Fig. 2h.

“h, Evaluation of ATc-induced growth inhibition by spotting a ten-fold serial dilution of cells on agar plates. **Representative images for one of the two tested clones are shown.**”

Ln 239-242: The authors present SEC data for purified Wzm-Wzt mutants to assess whether the engineered mutations affect complex folding/assembly. Did they by chance also assess whether both

subunits (Wzm and Wzt) are present in the peak fraction(s)? Because the complex is being purified via a tag on the transmembrane subunit, it is conceivable that deleting GH might lead to misfolding of the NBD but that the TMDs still dimerize and can be purified. Given the size of the TMDs plus detergent micelle, loss of the NBDs might only lead to a modest shift in retention time. Based on the traces in Sup Fig 6g, the main peak appears to elute slightly before 11 mL, while Δ GH appears to be shifted to ~12 mL. An assay like SDS-PAGE or Western blotting of the peak fractions would provide more clarity on whether both subunit subunits are present and could therefore be much more convincing.

We thank reviewer#3 for bringing up this justified concern. In fact, we analysed purified mutants by SDS-PAGE and confirmed presence of two bands, corresponding to Wzm and Wzt, for most of Wzm-Wzt variants. The representative SDS gels are included next to SEC profiles in Supplementary Fig. 6h. However, as can be seen on the respective gels, some variants gave rise to less pure material (as it was generally challenging to express and purify Wzm-Wzt_{Mabs}). Further, the TMD and the NBD run at similar heights, in particular in case of the Δ GH variant. Therefore, we performed a mass spectrometry analysis of the bands corresponding to Wzm-Wzt region from SDS-PAGE analysis of the individual fractions from SEC, which confirmed the presence of both Wzm and truncated Wzt (Δ GH) eluting at the volume corresponding to the Wzm-Wzt_{Mabs} complex. These data are presented in a new Supplementary Fig. 9.

Ln 301-305: See my comment on ligand modeling. I feel like this discussion of broad substrate specificity is a bit speculative given my concerns about the potential unreliability of ligand poses.

We respectfully disagree with reviewer#3 on this specific point. Our data clearly shows that the cavity can accommodate different ligands, including LMNG, lipids and the LLG analogue. We agree that the binding pose of the LLG analogue leaves room for interpretation (as addressed in greater details in the revised version of the manuscript). But the statements in line 301-305 do refer to the fact that this region of Wzm-Wzt is promiscuous in accepting different amphiphilic molecules. We think that this statement reflects pretty well what we saw in the different structures.

We appreciate the constructive feedback from the reviewers, and we have carefully addressed all points raised. Following editorial guidance, we have made substantial revisions to the manuscript to appropriately reflect the ambiguity in ligand assignment while still presenting our structural interpretations transparently.

Major changes made:

1. **Ligand deposition and nomenclature.** Following the editor's and Reviewer #3's suggestion, we have now deposited ligands in ADP-bound nanodisc structure using the "unknown ligand" (UNL) code in the PDB.
2. **Complete Spot Plate Documentation.** Following Reviewer #3's request, we repeated the spot plate assay for all mutants once more, spotting the strains on a single plate, arranged as they are shown in the figures.
3. **Revised language throughout manuscript.** We have systematically toned down definitive statements about ligand identity and binding poses.

Below, we provide a detailed response to each comment.

Reviewer #1 (Remarks to the Author):

The authors have addressed my previous critiques adequately. I have no further points to make and I believe the manuscript is ready for publication.

We thank Reviewer #1 for their positive assessment and for their constructive feedback during the first round of review, which substantially improved the manuscript.

Reviewer #2 (Remarks to the Author):

In this revised manuscript, the authors have addressed my concerns mostly around the cryoem structures and model interpretation. While I believe there is still some ambiguity over the interpretation, this has been acknowledged in the manuscript and alternative explanations proposed. One suggestion prior to publication would be to explain the lack of measured ATPase activity in either LMNG or nanodisc using the malachite green assay, which is at odds with the turn over conditions of some of the structures.

We thank Reviewer #2 for this suggestion. We indeed thought whether we should show the apparent lack of ATPase activity in our purified and nanodisc-reconstituted samples in an extra supplementary figure. However, since we observe both ATP-bound and ADP-bound conformations within the same turnover dataset for the wild-type transporter, but only the ATP-bound conformation for the ATPase-deficient E178Q mutant, our cryo-EM data actually demonstrates that ATP hydrolysis does occur. In our interpretation, the reason why we could not measure ATPase activity with the malachite green assay is that basal ATPase activity is likely extremely slow *in vitro*. By contrast, the cryo-EM analysis is

more sensitive to pick-up different conformations under ATP turnover than the traditional malachite green assay.

Reviewer #3 (Remarks to the Author):

I appreciate the authors detailed response to the issues I raised in my critique, and most of my concerns have been resolved. However, I feel that two significant issues are not adequately addressed.

1) The spot plate assays: I appreciate that the authors have now shown all of the plates to us reviewers, and everything appears to me to be adequately controlled. But unless I missed it, this data is still not in the manuscript for a reader to assess for themselves. This reduces the transparency and rigor of the final manuscript, as key controls are not being presented and preserved in the scientific record. I think the best solution is to repeat the experiments once more, spotting the strains on a single plate, arranged as the authors want to present them in the final figure. Then there need be no cropping, and there will be no question in a readers mind. Alternatively, and more complicated, would be to include all of the original plates (as provided in the response to reviewers document) in a supplementary figure, so that for each mutant/strain tested, a reader can see the controls from that experiment.

We agree with Reviewer #3 that transparency and rigor require showing complete experimental controls. Following the reviewer's suggestion, we have repeated the spot plate assays once more and included the new data in uncropped form in the Fig. 2h and Supplementary Fig. 7b.

2) Ligand modeling: Reviewer 2 and myself both raised similar concerns about the modeling of the ligands, and I still am concerned that one or more ligands may be modeled incorrectly. Why am I concerned? First, there is the question of the IDENTITY of the ligand(s) responsible for the observed density. This could be any molecule present in the cells the protein was isolated from; any component of the buffers used and anything else knowingly added to the sample; and also anything the sample was unintentionally exposed to (e.g., impurities in chemicals used, chemicals leaching from plastics used, etc). See PMID: 20944227. That is a lot of possibilities to narrow down, even if we just limit ourselves to considering cellular lipids, and anything the protein was exposed to in vitro. The density could result from the superposition of multiple things, not just one. Then there is the binding pose, which is also often tough at these sorts of resolutions. So agreement between the density and the ligand, and the chemical plausibility of the interaction are key factors. In this case, I feel both are in question. The discrepancy between the binding poses of the 3 different proposed ligands and the interactions between the ligands and the protein raise questions about chemical plausibility. The fact that the entire molecules don't fit into the observed density, and the modest complementarity between density and modeled ligands, raises questions about the whether the assignment of their identity is correct. Numerous molecules could be docked into this density with fits that are similar quality to the modeled LLG. That says that the map is not providing much information about the identity or pose of the the bound molecule(s). The authors may be right, perhaps part of LLG is disordered, and that is why the map can only explain a part of it; however, it seems very plausible to

me that it simply is not LLG, or the density is an average with contributions from LLG along with other lipid and detergent. I do not feel that the evidence is strong enough to assign the identify of these ligands, when they are a major crux of the manuscript. Perhaps these ligands should be deposited using the "unknown ligand" code UNL? Consequently, I still believe that speculating on the promiscuity of the ligand binding site lacks rigor, when the identity of the bounds ligands is in question--one can't begin to discuss how such diverse ligands are bound, if the evidence supporting their binding to a particular site with a particular pose is weak.

We thank reviewer #3 for expressing these relevant concerns. We agree with the reviewer that data interpretation is a somewhat delicate matter in this particular case, owing to the limited resolution and the fact that the extra density we observe does not fit the entire LLG molecule. Nevertheless, we feel that there appears to be a misunderstanding in relation of two concerns raised by the reviewer:

i) Doubts regarding the identity of the ligand:

We wish to emphasize again that the non-proteinaceous density we interpreted as LLG looks clearly different in the control sample, which was prepared without adding LLG. This data/evidence was presented already in the first submitted version of the manuscript and, in our view, this result excludes the possibility that *"this [density] could be any molecule present in the cells the protein was isolated from; any component of the buffers used and anything else knowingly added to the sample; and also anything the sample was unintentionally exposed to (e.g., impurities in chemicals used, chemicals leaching from plastics used, etc)."*

ii) Doubts regarding the chemical plausibility

As we discussed in the revised version in greater detail, there is no striking conflict concerning the chemical plausibility for any of the modeled ligands (i.e. LMNG, lipid or the two possible binding poses of LLG). At the same time, we agree that chemical plausibility does not support a specific LLG binding pose. The binding poses of LMNG and lipid are unambiguous in our view.

To address the remaining ambiguities better and to comply with editorial recommendations on how to present and discuss "difficult to interpret" non-proteinaceous densities, we made the following additional revisions:

1. Ligand deposition: Following the reviewer's suggestion, we have now deposited ligands in the ADP-bound nanodisc structure using the "unknown ligand" (UNL) code.
2. Revised language throughout: we have systematically toned-down definitive statements about ligand identity and binding poses.

Specific changes are:

We removed any statement about the LLG bound structure in the abstract.

Line 117-119 (previously): "but we only observed density attributable to this substrate when the transporter was prepared in nanodiscs"

Revised to: "but we only observed unassigned density consistent with an amphipathic molecule in the binding cavity when the transporter was prepared in nanodiscs in the presence of the substrate analog"

Line 263-270 (previously definitive statements about farnesyl-P-P-GlcNAc binding):

Revised to: " Importantly, we also prepared a control sample without adding the substrate analog (see section below and Fig. 4). Cryo-EM analysis of the control sample revealed a different unassigned density in a similar but distinct location, which we interpreted as a phospholipid. This suggests that the density observed in the substrate analog-added sample most likely corresponds to farnesyl-P-P-GlcNAc. Therefore, we modeled this substrate analog into the density with the farnesyl moiety entering first (termed "lipid-first" binding pose) as one possible interpretation (Fig. 3c). However, we acknowledge that the moderate resolution and incomplete density coverage leave room for alternative interpretations."

Line 274-275 (previously): "The cryo-EM density can also be interpreted in an alternative way wherein the Galf-Galf-Rhap-GlcNAc-P-P-moiety of the substrate analog enters first ("sugar-first" binding pose) (Supplementary Fig. 12)"

Revised to: "The observed density could represent the substrate analog in an alternative 'sugar-first' orientation wherein the Galf-Galf-Rhap-GlcNAc-P-P-moiety enters first (Supplementary Fig. 12)."

Line 313-315 (PE lipid section, previously): "The shape resembles a lipid and hence we built the abundant E. coli lipid phosphatidylethanolamine (PE) into the density (Fig. 4b)."

Revised to: "The shape resembles a lipid and hence we built the abundant E. coli lipid phosphatidylethanolamine (PE) into the density as one of the possible interpretations (Fig. 4b)."

Lines 322-325 (PE lipid section, previously): "In summary, our cryo-EM structures revealed that the chemistry and geometry of the TMD cavity can accommodate molecules with diverse biophysical properties, including the lipid moiety of farnesyl-P-P-GlcNAc-Rhap-Galf-Galf, the head group of a PE lipid, and the maltose group of LMNG (Supplementary Fig. 15)."

Revised to: "In summary, our cryo-EM structures revealed that the chemistry and geometry of the TMD cavity can accommodate different amphipathic molecules: the detergent LMNG, a phospholipid,

and a density we interpreted as the LLG substrate analog farnesyl-P-P-GlcpNAc-Rhap-Galf-Galf(Supplementary Fig. 15).”

We are grateful to the reviewers for their time, expertise, and thoughtful evaluation of our work.

Reviewer #2 (Remarks to the Author):

The acknowledgement of the ambiguous quality of the ligand density and marking of the PDB as such (UNL) is appreciated and important for future users of the coordinates. Not clear that the timeframe of the cryoEM incubation prior to vitrification and the assay conditions for the malachite green assay are so different that one would show turnover and the other not. However I am willing to accept the paper with the current revisions.

We appreciate the reviewer's acknowledgment of our transparent reporting regarding the ligand density quality and the UNL designation in the PDB deposition.

Regarding the reviewer's comment about the timeframe differences between cryo-EM sample preparation and the malachite green assay conditions, we would like to clarify how we think this apparent contradiction can be resolved:

We actually do not think that it is a matter of the timeframe (which is indeed not really different between grid preparation and the malachite green assay), but the apparent discrepancy can be explained by the different readouts of the assays.

ATP hydrolysis by purified Wzm-Wzt appears to be very slow and consequently, free phosphate does not accumulate to detectable levels and therefore cannot be measured by the malachite green assay. However, slow ATP hydrolysis does not inform on the thermodynamic equilibrium of ATP-bound versus ADP-bound transporter in the presence of ATP. In fact, in our cryo-EM analyses, we observe both ATP-bound and ADP-bound states in the presence of ATP for the wild-type transporter, while the Walker B EtoQ mutant is trapped in the ATP-bound state in the presence of ATP. From these observations, we conclude that the wild-type transporter must exhibit (slow) ATPase hydrolysis, because otherwise it would not be possible to observe both conformational states by cryo-EM. Hence, in this particular case of very slow ATPase activity, cryo-EM is more sensitive in detecting ATP hydrolysis, because it infers it by analyzing the distribution of conformational states.

Reviewer #3 (Remarks to the Author):

I appreciate the authors responsiveness to my concerns. They are now resolved and I strongly support acceptance. Nice manuscript!

We are pleased that all concerns have been resolved and appreciate the positive assessment of our manuscript.